# Online single-cell data integration through projecting heterogeneous datasets into a common cell-embedding space

Lei Xiong [1,2,3,8], Kang Tian [1,2,8], Yuzhe Li[1,4], Weixi Ning[1], Xin Gao [5,6,7] & Qiangfeng Cliff Zhang [1,2] ✉

Computational tools for integrative analyses of diverse single-cell experiments are facing formidable new challenges including dramatic increases in data scale, sample heterogeneity, and the need to informatively cross-reference new data with foundational datasets. Here, we present SCALEX, a deep-learning method that integrates single-cell data by projecting cells into a batch-invariant, common cell-embedding space in a truly online manner (i.e., without retraining the model). SCALEX substantially outperforms online iNMF and other state-of-the-art non-online integration methods on benchmark single-cell datasets of diverse modalities, (e.g., single-cell RNA sequencing, scRNA-seq, single-cell assay for transposase-accessible chromatin use sequencing, scATAC-seq), especially for datasets with partial overlaps, accurately aligning similar cell populations while retaining true biological differences. We showcase SCALEX's advantages by constructing continuously expandable single-cell atlases for human, mouse, and COVID-19 patients, each assembled from diverse data sources and growing with every new data. The online data integration capacity and superior performance makes SCALEX particularly appropriate for large-scale single-cell applications to build upon previous scientific insights.

Single-cell experiments enable the decomposition of samples into their constituent, diverse cell-types and cell states[1–4]. Many computational tools have been developed for integrative analysis of single-cell datasets, all seeking to separate biological variations from non-biological noise, such as batch effects of different donors, conditions, and/or analytical platforms[5,6]. The scope of the integration task is expanding rapidly with technical advances for single-cell studies, which continue to grow larger and larger in scale, now exceeding 1 million cells in some cases[7,8]. Moreover, the range of examined sample types is also increasing, and datasets now often include highly heterogenous cell subsets[9,10]. Most importantly, as single-cell studies become more routine, new studies should be informatively cross-referenced to foundational research stuides[7,8,11–15]. Thus, there is a growing need for integration tools that can manage single-cell data of large-scale and complex cell-type compositions while also supporting accurate alignment to and exploration within existing datasets.

Most current single-cell data integration methods (e.g., Seurat[16–18], MNN[19], Harmony[20], Conos[21], Scanorama[22], BBKNN[23], etc.) are based on

[1]MOE Key Laboratory of Bioinformatics, Beijing Advanced Innovation Center for Structural Biology & Frontier Research Center for Biological Structure, Center for Synthetic and Systems Biology, School of Life Sciences, Tsinghua University, Beijing 100084, China. [2]Tsinghua-Peking Center for Life Sciences, Beijing 100084, China. [3]Shanghai Qi Zhi Institute, Shanghai 200030, China. [4]Academy for Advanced Interdisciplinary Studies, Peking University, Beijing 100871, China. [5]Computer Science Program, Computer, Electrical and Mathematical Sciences and Engineering (CEMSE) Division, King Abdullah University of Science and Technology (KAUST), Thuwal 23955-6900, Kingdom of Saudi Arabia. [6]KAUST Computational Bioscience Research Center (CBRC), King Abdullah University of Science and Technology (KAUST), Thuwal 23955-6900, Kingdom of Saudi Arabia. [7]BioMap, Beijing 100086, China. [8]These authors contributed equally: Lei Xiong, Kang Tian. ✉e-mail: qczhang@tsinghua.edu.cn

the searching across batches for cell-correspondence, for instance similar individual cells or cell anchors/clusters. These methods suffer from three limitations. First, they are prone to mixing cell populations that only exist in some batches, which becomes a severe problem for the integration of complex datasets that contain non-overlapping cell populations in each batch (i.e., partially overlapping data)[16,17]. Second, they require computational resources that increase dramatically as the number of cells and of batches increase, making these methods increasingly unsuitable for today's large-scale single-cell datasets[7,8,11–15]. Finally, these methods can only remove batch effects from the current dataset being assessed. Each time a new dataset is added, it requires an entirely new integration process that changes the existing integration results of previous studies. This requirement severely limits a tool's ability to continuously integrate arriving new single-cell data without recalculating existing integrations from scratch, a capacity referred to as "online" data integration[24].

Online data integration ability is becoming increasingly crucial with today's single-cell experiments. The recently developed tool, online iNMF[24], an online version of LIGER[25], iteratively applies integrative non-negative matrix factorization (iNMF) to decouple the shared and dataset-specific factors related to cell identities, and thus is able to incorporate new data with existing datasets on-the-fly. Another recently developed package, scvi-tools[26], combining scVI[27] with scArches[28], applies a conditional variational autoencoder (VAE)[29] framework to model the inherent distribution of the input single-cell data for data integration. However, the conditional VAE design of scVI requires model augmentation and retraining when integrating new data, meaning that scVI is not an online method. We want to highlight that this online integration ability meets a rapidly growing need in the life sciences and in biomedicine: it enables the alignment of data coming from new single-cell analyses (from the lab and clinic) into the substantial corpus of existing knowledge, especially that from previous foundational single-cell research. Put another way, the online integration capacity obviates the need to augment and/or retrain models when analyzing additional datasets, which both preserves hard-won scientific insights and saves a huge amount of computational resource.

Here, we developed SCALEX as a method for online integration of heterogeneous single-cell data based on a VAE framework. The encoder of SCALEX is designed to be a data projection function that only preserves batch-invariant biological data components when projecting single-cells. Importantly, the projection function is a generalized one that requires no retraining on new data, thus allowing SCALEX to integrate single-cell data in an online manner. Working with an extensive collection of benchmark datasets, we demonstrate that SCALEX substantially outperforms online iNMF as well as non-online single-cell data integration tools, in terms of integration accuracy, scalability, and computationally efficiency. The advantages make SCALEX particularly appropriate for the integration and research utilization of today's single-cell datasets, which continue to grow along with the ongoing explosion of single-cell studies in biology and medicine.

## Results

### SCALEX implements a generalized encoder that enables online integration of single-cell data

To enable online integration, the fundamental design concept underlying SCALEX is to implement a *generalized* projection function that disentangles the batch-related components away from the batch-invariant components of single-cell data and projects the batch-invariant components into a common cell-embedding space. We previously applied VAE and designed SCALE (Single-Cell ATAC-seq Analysis via Latent feature Extraction) to model and analyze single-cell ATAC-seq data[30]. We found that the encoder of SCALE has the potential to disentangle cell-type-related and batch-related features in a low-dimensional embedding space.

Here, to obtain a generalized encoder for data projection without retraining, SCALEX includes three specific design elements (Fig. 1a, Supplementary Fig. 1, "Overview of the SCALEX model" in Methods). First, SCALEX implements a batch-free encoder that extracts only biological-related latent features (**z**) from input single-cell data (**x**) and a batch-specific decoder[29] that reconstructs the original data from **z** by incorporating batch information back during data reconstruction. Supplying batch information only to the decoder focuses the encoder exclusively on learning the batch-invariant biological components, which is crucial for the encoder generalizability. In contrast, scVI includes a set of batch-conditioned parameters into its encoder, which restrains the encoder from the generalizability with new batches and thus precludes online data integration. Second, SCALEX includes a Domain-Specific Batch Normalization (DSBN)[31] layer using multi-branch Batch Normalization[32] in its decoder to support incorporation of batch-specific variations during single-cell data reconstruction. Third, the SCALEX encoder employs a mini-batch strategy that samples data from all batches (instead of a single batch), which more tightly follows the overall distribution of the input data. Note that each mini-batch is subjected to a Batch Normalization layer in the encoder to adjust the deviation of each mini-batch and to align it to the overall input distribution.

We conducted extensive analyses of SCALEX hyperparameters and also tested the specific contributions of each design element by implementing a set of SCALEX test-variants, each lacking an individual or a combination of the design elements, and evaluating their performance for single-cell data integration ("Ablation studies using test-variants of SCALEX" in Methods). We found that each design element is crucial for the integration performance of SCALEX. More importantly, the combination of these design elements renders the encoder of SCALEX a generalized function capable of accurate projection of single cell data from different batches into a batch-invariant cell-embedding space, making SCALEX a truly online data integration method.

### SCALEX integration is substantially more accurate than state-of-the-art single-cell data integration methods

We extensively assessed the basic data integration performance of SCALEX, following the evaluative framework proposed in a recent comparative study[33]. We examined multiple well-curated scRNA-seq datasets, including human *pancreas* (eight batches of five studies)[34–38], *heart* (two batches of one study)[39] and *liver* (two studies)[40,41]; as well as human non-small-cell lung cancer (*NSCLC*, four studies)[42–45] and peripheral blood mononuclear cells (*PBMC*; two batches assayed by two different protocols)[16]. Our comparison included online iNMF and other state-of-the-art non-online single-cell data integration methods, including Seurat v3, Harmony, MNN, Conos, BBKNN, Scanorama, LIGER (i.e., batch iNMF), and scVI. We evaluated the integration performance of these tools based on the benchmark datasets by Uniform Manifold Approximation and Projection (UMAP)[46] embedding visualization as well as a series of scoring metrics[19,20,47–49].

With UMAP embedding, we note that all of the raw datasets displayed strong batch effects, with cell-types that were common in different batches separately distributed. Overall, SCALEX, Seurat v3, and Harmony achieved the best integration performance for most of the datasets by merging common cell-types across batches while keeping disparate cell-types apart (Supplementary Fig. 2). MNN, scVI, and Conos integrated many datasets but left some common cell-types not well-aligned. Online iNMF, LIGER, BBKNN, and Scanorama often had unmerged common cell-types, and sometimes incorrectly mixed distinct cell-types together. For example, considering the T cell populations between the two batches in the *PMBC* dataset (Fig. 1b), while SCALEX, Seurat v3, Harmony, MNN, scVI integrations were effective, online iNMF misaligned some of the CD4 naïve T cells with CD8 naïve T cells, and misaligned some NK cells with CD8 T cells.

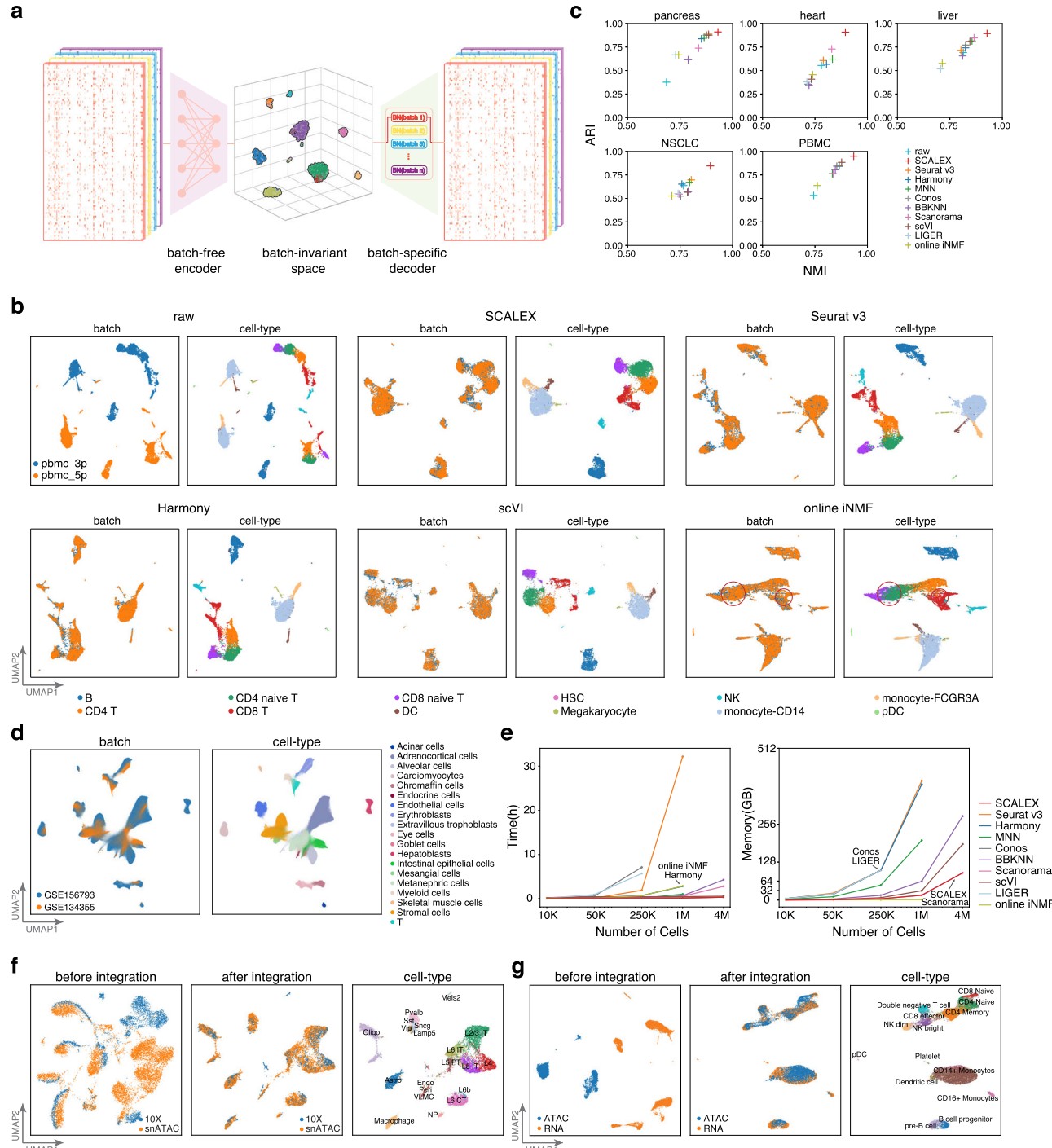

**Fig. 1 | The design and performance of SCALEX for single-cell data integration.** **a** SCALEX models the global structure of single-cell data using a variational auto-encoder (VAE) framework. **b** UMAP embeddings of the *PBMC* dataset before and after integration by indicated methods. Cells are colored by batch (left) and cell-type (right). Misalignments are highlighted with red circles. **c** Scatter plot comparing SCALEX and the other state-of-the-art single-cell data integration tools in terms of the ARI score (y-axis) and the NMI score (x-axis), based on the Leiden clustering results in the latent space across the indicated benchmark datasets. **d** UMAP embeddings of the SCALEX integration of the Human Fetal Atlas dataset after integration by SCALEX, colored by batch and cell-type. **e** Comparison of computation efficiency based on datasets of different sizes sampled from the whole Human Fetal Atlas dataset) including runtime (left) and memory usage (right). Online iNMF was not successfully tested on 4 M data due to a HDF5 file conversion issue for large data ("Online iNMF and LIGER (LIGER, v1.0.0)" subsection of "Comparison with other integration methods" in Methods). **f** UMAP embeddings of the mouse brain scATAC-seq dataset before (left) and after integration (middle, right); colored by data batch or cell-types. **g** UMAP embeddings of the PBMC scRNA-seq and scATAC-seq cross-modality dataset before (left) and after SCALEX integration (middle, right); colored by batch or cell-type.

SCALEX substantially outperformed all of the other methods for cell-type clustering, as assessed by the adjusted Rand Index (ARI)[47] and the Normalized Mutual Information (NMI)[48] (Fig. 1c, Supplementary Figs. 3, 4). To quantify cell-type separation and batch mixing we used two paired metrics: a pair comprising the Silhouette score[49] and the batch entropy mixing score[19], as well as a pair comprising the cell-type and integration local inverse Simpson's Indexes (cLISI and iLISI)[20]. Overall, SCALEX achieved

the highest scores for cell-type separation, and tied with Seurat v3 and Harmony as the best-performing methods on the batch mixing metrics (Supplementary Fig. 4a). Interestingly, we observed that both LIGER and online iNMF often scored the lowest for cell-type separation yet the highest for batch mixing. However, after careful investigation, we concluded that a higher batch mixing score does not necessarily indicate better data integration, but instead often indicates an issue of over-correction, which we consider in-depth in a dedicated subsection below. Finally, we followed the protocol in a recent large-scale study for benchmarking single-cell integration methods to compare SCALEX against ten state-of-the-art methods using multiple scores[50] ("Single-cell integration benchmarking (scIB)" in Methods). We observed that SCALEX outperformed all other tools on the *pancreas*, *liver*, and *NSCLC* datasets in terms of the overall score, and ranked the third on the *PBMC* dataset and the fourth on the *heart* dataset (Supplementary Fig. 4b).

### SCALEX is scalable to Atlas-level datasets and accommodates diverse data modalities

Single-cell datasets that contain a large number of cells and consist of heterogenous and complex samples from multiple tissues have been termed "Atlas-level" datasets in a recent comparative study[33]. These Atlas-level datasets are posing new challenges to data integration tools. We tested the scalability and computation efficiency of SCALEX by applying it to a typical Atlas-level dataset, the Human Fetal Atlas dataset, which contains 4,317,246 cells from two data batches, GSE156793 and GSE134355 (Supplementary Fig. 5a, b, "Preprocessing for scRNA-seq" in Methods)[8,15]. SCALEX accurately integrated these two batches, showing good alignment of the same cell-types (Fig. 1d). In addition to SCALEX, only BBKNN, Scanorama, and scVI can be used to integrate this Atlas-level dataset, however, their integrations does not separate and align the cell-types well, as indicated by the UMAP embeddings (Supplementary Fig. 5c) and the low cell-type separation and batch mixing scores (Supplementary Fig. 5d).

We compared the computational efficiency of different methods using down-sampled datasets (of 10 kilo (K), 50 K, 250 K, 1 million (M), and 4 M cells) from this Human Fetal Atlas dataset. Both SCALEX and online iNMF consumed very efficient runtime and memory that increased only linearly with data size. scVI also is scalable to 4 M cells with acceptable memory usage, whereas Seurat v3, Harmony, Conos, and LIGER consumed runtime and/or memory that increased exponentially, thus did not scale beyond 1 M cells on a workstation of 64 central processing unit (CPU) cores and 256 gigabytes (GB) memory (Fig. 1e). Notably, the deep learning framework of SCALEX enables it to run very efficiently on graphics processing unit (GPU) devices, requiring much reduced runtime (using about 20 minutes and 90 GB of memory on the 4 M dataset).

SCALEX can be used to integrate other modalities of single-cell data (e.g., scATAC-seq[51,52], cellular indexing of transcriptomes and epitopes by sequencing, CITE-Seq[53], etc.) and cross-modality data (e.g., simultaneous analysis of scRNA-seq and scATAC-seq). SCALEX substantially outperformed all other methods for integration of mouse brain scATAC-seq datasets (two batches assayed by single nucleus assay for transposase-accessible chromatin using sequencing, snATAC and 10X)[54] (Fig. 1f, Supplementary Fig. 6a–c), and performed well for integration of additional single-cell data modalities including CITE-seq[53] and spatial transcriptome MERFISH data[55] (Supplementary Fig. 6d, e). We also used SCALEX to integrate a cross-modality dataset (scRNA-seq and scATAC-seq)[56,57] and found that SCALEX correctly integrated the two modalities of data and distinguished rare cells that are specific to the scRNA-seq data, including pDC and platelet cells (Fig. 1g), doing so better than other methods including two additional methods scjoint[58] and bindSC[59], according to both UMAP embeddings and multiple analytical metrics (Supplementary Fig. 7).

### SCALEX integrates partially overlapping datasets without over-correction

Many recent single-cell datasets, especially Atlas-level datasets, feature high sample heterogeneity and complex cell-type compositions[9,10]. These datasets often contain partially overlapping batches where each batch contains some non-overlapping cell populations. For example, the *liver* dataset is a partially overlapping dataset where the hepatocyte population contains multiple subtypes specific to different batches: three subtypes are specific to LIVER_GSE124395, and two other subtypes only appear in LIVER_GSE115469 (Supplementary Fig. 8).

This partial overlap problem presents a major challenge for single-cell data integration and often leads to an issue of over-correction (i.e., mixing of distinct cell-types), especially for those local cell similarity-based methods[16,17]. For example, Seurat v3 mixed the hepatocyte-CXCL1, hepatocyte-CYP2A13, and hepatocyte-TAT-AS1 cells and Harmony mixed the hepatocyte-CYP2A13 and hepatocyte-TAT-AS1 cells (Fig. 2a). As a global integration method that projects cells into a common cell-embedding space, SCALEX is expected to be less sensitive to this problem. Indeed, we noticed that SCALEX correctly maintained the five hepatocyte subtypes apart (as did scVI. Fig. 2a). Unexpectedly, despite being a global method, online iNMF severely suffered from over-correction, mixing all five hepatocyte subtypes, and even mixing B cells and NK cells (Fig. 2a), presumably because its matrix factoring algorithm forced the alignment of distinct cell-types.

We defined an over-correction score, a metric to measure this over-correction problem based on the percent of cells with inconsistent cell-types in the neighborhood for each cell ("Over-correction score" in Methods). Formally, the over-correction score is a negative index, i.e., the higher the over-correction score, the more severe the extent of inaccurate mixing of cell-types. For the benchmark datasets, SCALEX had the lowest over-correction scores (Fig. 2b), whereas online iNMF yielded extremely high over-correction scores.

To systematically characterize the performance of different methods on partially overlapping datasets, we constructed test datasets with a range of common cell-types, that we generated based on down-sampling of the six major cell-types in the *pancreas* dataset ("Generation of partially overlapping datasets" in Methods). SCALEX integration was accurate for all cases, aligning the same cell-types without over-correction, whereas Seurat v3, Harmony, and online iNMF frequently mixed distinct cell-types (Fig. 2c, d). Although scVI showed one of the lowest levels of over-correction when integrating partially overlapping datasets, it is prone to mistakenly splitting one cell-type into many small groups. We noted that the severity of over-correction and error-splitting is amplified as the overlapping number decreases (Supplementary Fig. 9). When there were no common cell-types, both Seurat v3 and Harmony collapsed the six cell-types into three, mixing alpha with gamma cells, beta with delta cells, and acinar with ductal cells to varying extents, whereas scVI split alpha cells into 6 groups. We repeated this down-sampling analysis from the 12 cell-types in the *PBMC* dataset and observed similar results of over-correction and error-splitting (Supplementary Fig. 10).

### SCALEX increases the scope and resolution of an existing cell space by adding new data through online projection

The generalizability of SCALEX's encoder to project cells from various sources into a common cell-embedding space without model retraining allows SCALEX to integrate new single-cell data with existing data in an online manner. We tested the online data integration performance of SCALEX for newly arriving data based on the *pancreas* dataset. Prior to projection, we first used SCALEX to integrate the *pancreas* dataset and this accurately removed the strong batch effect that was evident in the raw data (Fig. 3a, Supplementary Fig. 11a, b).

We subsequently projected three new batches of scRNA-seq data[60–62] for pancreas tissues (Fig. 3b) into this "pancreas cell space" using the same SCALEX encoder trained on the original *pancreas*

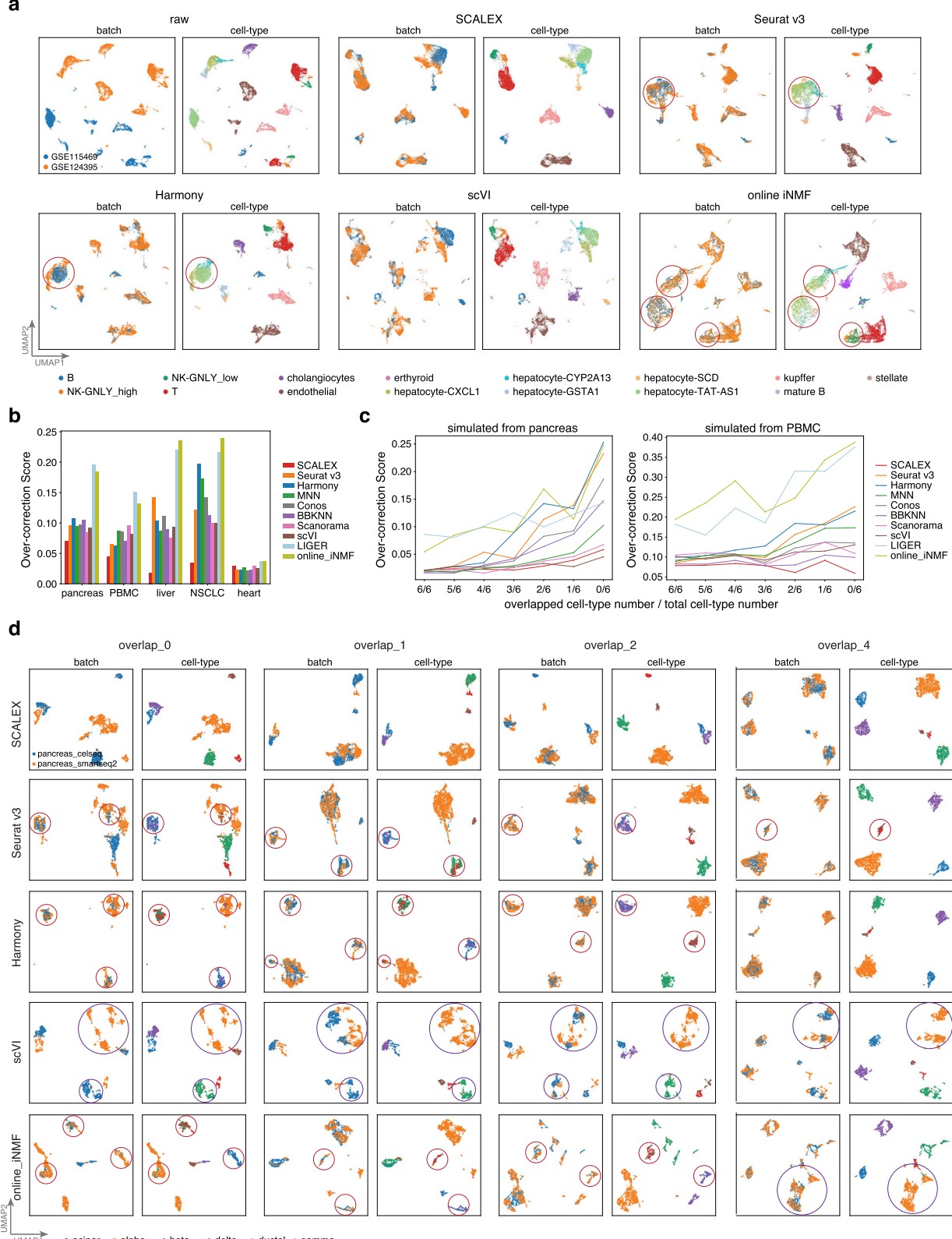

**Fig. 2 | Comparison of integration performance over partially overlapping datasets by different methods. a** UMAP embeddings of the integration results by the indicated methods based on the *liver* dataset. **b** Over-correction score of different methods based on the indicated benchmark datasets. **c** Over-correction score of the indicated methods based on the simulated datasets, with decreased numbers of common cell-types (obtained by down-sampling the *pancreas* and *PBMC* dataset). **d** UMAP embeddings of the integration results by the indicated methods based on the simulated *pancreas* datasets with different numbers of specified common cell-types. Over-corrections are highlighted with red circles and error-splittings with purple circles, respectively.

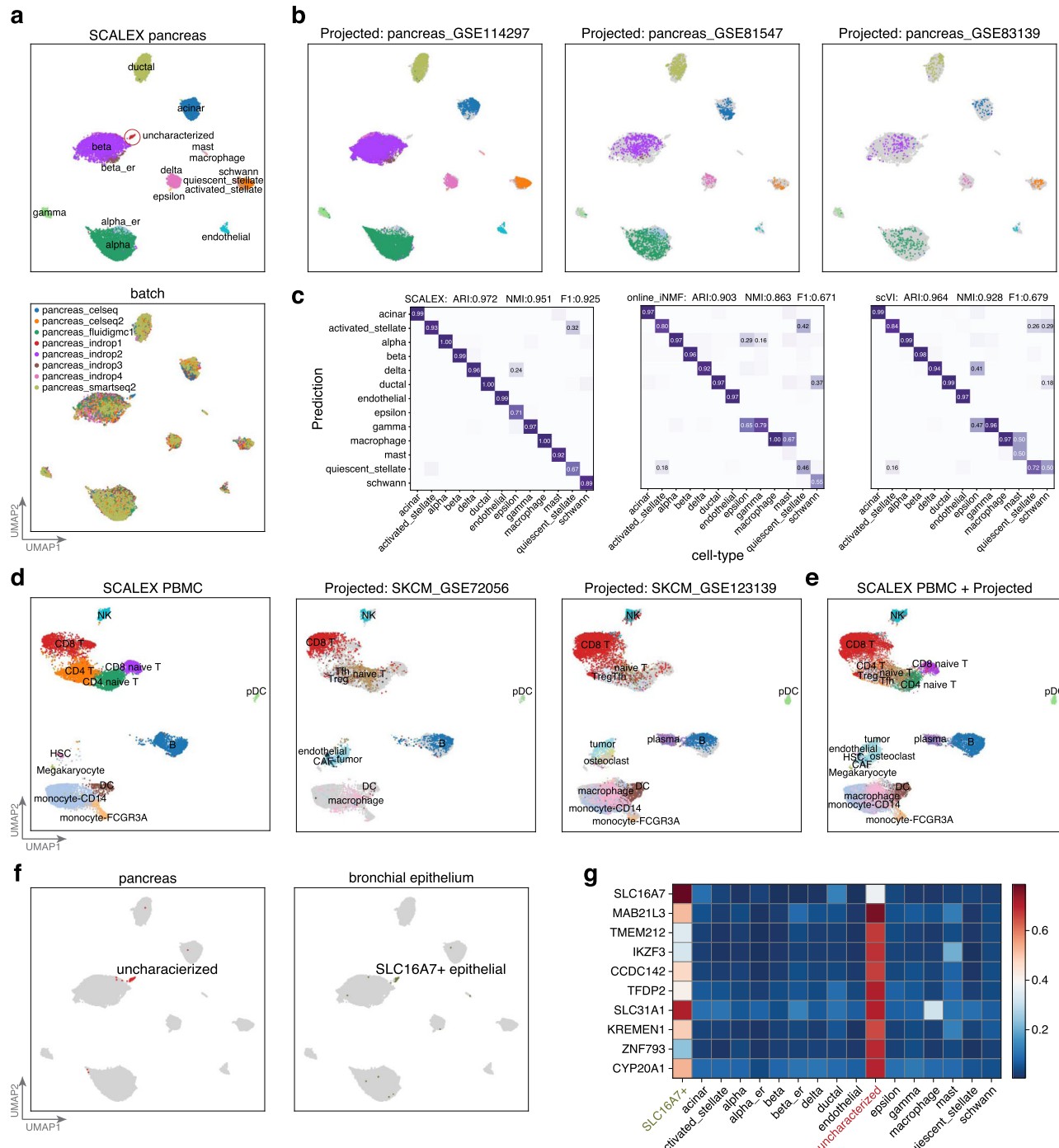

**Fig. 3 | Projecting heterogenous data into a common cell-embedding space.**
**a** UMAP embeddings of the *pancreas* dataset after integration by SCALEX, colored by cell-type and by batch. **b** UMAP embeddings of the common cell space obtained by using SCALEX to project three additional indicated *pancreas* data batches onto the *pancreas* dataset. Cells are colored by cell-type with light gray shadows representing the original *pancreas* dataset. **c** Confusion matrix between ground truth cell-types and those annotated by different methods. ARI, NMI and F1 scores (top) measure the annotation accuracy. **d** UMAP embeddings of the common space obtained by using SCALEX to project the two projected melanoma data batches onto the *PBMC* dataset, colored by cell-types with light gray shadows represent the original *PBMC* dataset. **e** UMAP embeddings of the common cell space that includes the original *PBMC* dataset and the two projected melanoma data batches.
**f** Annotating an uncharacterized small cell population in the *pancreas* dataset by projection of the bronchial epithelium data batches into the pancreas cell space. Only the uncharacterized cells in the *pancreas* dataset (left) and the SLC16A7+ epithelial cells in the bronchial epithelium data batches (right) are colored.
**g** Heatmap showing the normalized expression of the top-10 ranking specific genes for the uncharacterized cell population in different cell-types.

dataset. After projection, most of the cells in the new batches were accurately aligned to the correct cell-types in the pancreas cell space, enabling their accurate annotation by cell-type label transfer (Fig. 3c, "Cell-type annotation by label transfer" in Methods). We benchmarked projection accuracy by calculating the ARI, the NMI, and the F1 scores

to evaluate cell-type annotation by label transfer with cell-type information in the original studies. We compared the results with online iNMF and scVI, the only two tools that are able to project cells into an *existing* cell-space (note that data projection of scVI needs model retraining through scArches). SCALEX achieved the highest projection

accuracy in comparisons with online iNMF and scVI (Fig. 3c). scVI also achieved high accuracy, projecting most cells onto right locations, with only a few exceptions of alpha and ductal cells (Supplementary Fig. 11c). Online iNMF mixed distinct cell-types when incorporating new batches, e.g., projecting some alpha cells onto the locations of gamma and delta cells (Supplementary Fig. 11c), which in turn led to wrong annotations during label transfer (Fig. 3c).

The ability to project new single-cell data into an existing cell-embedding space allows SCALEX to readily enrich (i.e., to add biological resolution) this cell space with additional informative details. To verify this, we projected two additional melanoma data batches (SKCM_GSE72056, SKCM_GSE123139)[10,63] onto the previously constructed PBMC space. Again, SCALEX correctly projected all common cell-types onto the same locations in the PBMC cell space (Fig. 3d), but online iNMF mixed tumor cells with plasma, monocyte and CD8 T cells, and scVI split the CD8T cells into several distinct groups (Supplementary Fig. 12). Importantly, we noticed that for the tumor and plasma cells only present in the melanoma data batches, SCALEX did not project these cells onto any existing cell populations in the PBMC space; rather, it projected them onto new locations close to similar cells, with the plasma cells projected to a location near B cells, and the tumor cells projected to a location near HSC cells (Fig. 3e). This indicates that SCALEX can enrich an existing cell space with new cell-types through data projection.

SCALEX projection also enables *post hoc* annotation of unknown cell-types in an existing cell space using new data. For instance, we noted a group of previously uncharacterized cells in the *pancreas* dataset (Fig. 3a). We found that these cells displayed high expression levels of known epithelial gene markers. We therefore assembled a collection of epithelial cells from the *bronchial epithelium* dataset[64], and then projected these epithelial cells onto the pancreas cell space. We found that a group of antigen-presenting airway epithelial (SLC16A7 + epithelial) cells were projected onto the same location of the uncharacterized cells (Fig. 3f). These data, together with the observation that both cell populations showed similar marker gene expression (Fig. 3g), suggest that these uncharacterized cells are also SLC16A7 + epithelial cells. Note that online iNMF and scVI were not able to identify this small group of epithelial cells, because they were split into several smaller groups and/or were often mixed with other cell-types (Supplementary Fig. 2). SCALEX thus enables discovery science in cell biology by supporting exploratory analysis with large numbers of diverse datasets.

## SCALEX integration constructs expandable single-cell atlases
The ability to combine heterogenous data into a common cell-embedding space makes SCALEX a powerful tool to construct a single-cell atlas from a collection of diverse datasets. We applied SCALEX integration to three large and complex datasets: the Mouse Atlas dataset (comprising multiple organs from two studies assayed by 10X, Smart-seq2, and Microwell-seq[12,14]), the Human Atlas dataset (comprising multiple organs from two studies assayed by 10X and Microwell-seq[15,65]), and the Human Fetal Atlas dataset[8,15] (Supplementary Fig. 13).

Despite the strong batch effects in the raw data, SCALEX accurately integrated the three batches of the Mouse Atlas data into a common cell-embedding space (Fig. 4a–c, Supplementary Fig. 14a). Common cell-types were well-aligned at the same position in the cell space, including B, T, and endothelial cells presented in all tissues, and proximal tubule, urothelial, and hepatocytic cells from particular tissues. Distinct cell-types were located separately, such as sperm, Leydig, and small intestine cells from the Microwell-seq data, keratinocyte stem cells and large intestine cells from the Smart-seq2 data, indicating that biological variations were well preserved (Supplementary Fig 14b, c). We compared SCALEX with all other methods and found that SCALEX performed the best for cell-type clustering, especially for avoiding over-correction (Fig. 4d, e, Supplementary Fig. 13b).

Importantly, atlases generated with SCALEX can be further expanded by projecting new single-cell data to support comparative studies of cells both in the original atlas and in the new data. To illustrate this utility, we projected two additional data batches of aged mouse tissues from *Tabula Muris Senis* (Smart-seq2 and 10X)[13] and two single tissue datasets (lung and kidney)[66] onto the SCALEX Mouse Atlas cell space. We found that cells in the new data batches were correctly projected onto the locations of the same cell-types in the cell-embedding space of the initial atlas (Fig. 4f) as confirmed by the accurate cell-type annotations for the new data by label transfer (Fig. 4g).

Following the same strategy, we constructed a SCALEX Human Atlas by integration of multiple tissues from two studies (GSE134355, GSE159929) (Supplementary Fig. 15a, b). SCALEX effectively eliminated the batch effects in the original data and integrated the two datasets (Supplementary Fig. 15c, d). Again, we were able to correctly project two additional human skin datasets (GSE130973, GSE147424)[67,68] onto the Human Atlas cell space (Supplementary Fig. 15e), and accurately annotated these projected skin cells (Supplementary Fig. 15f). In sum, these results illustrate that SCALEX enables: i) researchers to evaluate their project-specific single cell datasets by leveraging existing information in large-scale (and ostensibly well annotated) cell atlases; and ii) atlas creators to informatively integrate new datasets and derive new biological insights from new research programs.

## An integrative SCALEX COVID-19 PBMC Atlas revealed distinct immune responses among COVID-19 patients
Many single-cell studies have been conducted to analyze COVID-19 patient immune responses[69–76]. However, these studies often suffer from small sample size and/or limited sampling of various disease states[70,76]. For a comprehensive study, we used SCALEX to generate a COVID-19 PBMC Atlas, integrating data from nine COVID-19 studies, involving a total of 860,746 single cells in 10 batches[69–75] (Fig. 5a, Supplementary Dataset 1). We identified 22 cell-types, each of which has support from gene expression data for canonical markers (Fig. 5b, c, Supplementary Fig. 16a, "Cell-type annotation by clustering" in Methods). Cells across different studies were integrated accurately with the same cell-types aligned together, confirming the integration performance of SCALEX (Supplementary Fig. 16b), which was much better than the other methods (Supplementary Fig. 16c, d).

Interestingly, we found that some cell subpopulations were differentially associated with patient status (Fig. 5d). A subpopulation of CD14 monocytes (CD14-ISG15-Mono) was characterized by its high expression of Type I interferon-stimulated genes (ISGs) and genes enriched with immune-response-related gene ontology (GO) terms (Fig. 5e, f). The frequency of CD14-ISG15-Mono cells increased significantly from mild/moderate to severe patients (Fig. 5g, Supplementary Fig. 17a, "Analysis of changes in cell-type frequency across multiple conditions" in Methods). Within the COVID-19 patients, we observed a significant decrease in ISG gene expression in CD14-ISG15-Mono cells between the mild/moderate and severe cases, suggesting an immune exhaustion-like response in severe COVID-19 patients[69] (Fig. 5e).

Additionally, a neutrophil subpopulation (NCF1-Immature-Neutrophil), characterized by decreased expression of the genes responsible for neutrophil activation but elevated expression of genes enriched with viral-process-related GO terms, was specifically enriched in severe verse mild/moderate patients (Supplementary Fig. 17b, c). A plasma cell subpopulation (MZB1-Plasma), characterized by decreased expression of genes related to antibody production and enriched for immune and inflammatory response-related GO terms, were also enriched in severe patients (Supplementary Fig. 17d, e). Thus, the SCALEX COVID-19 PBMC atlas, generated by integrating a highly diverse collection of single-cell data from individual studies, identified multiple immune cell-types that become progressively dysfunctional during COVID-19 disease progression[74]. Importantly, these cell trends were not and could not have been detected in the small-scale,

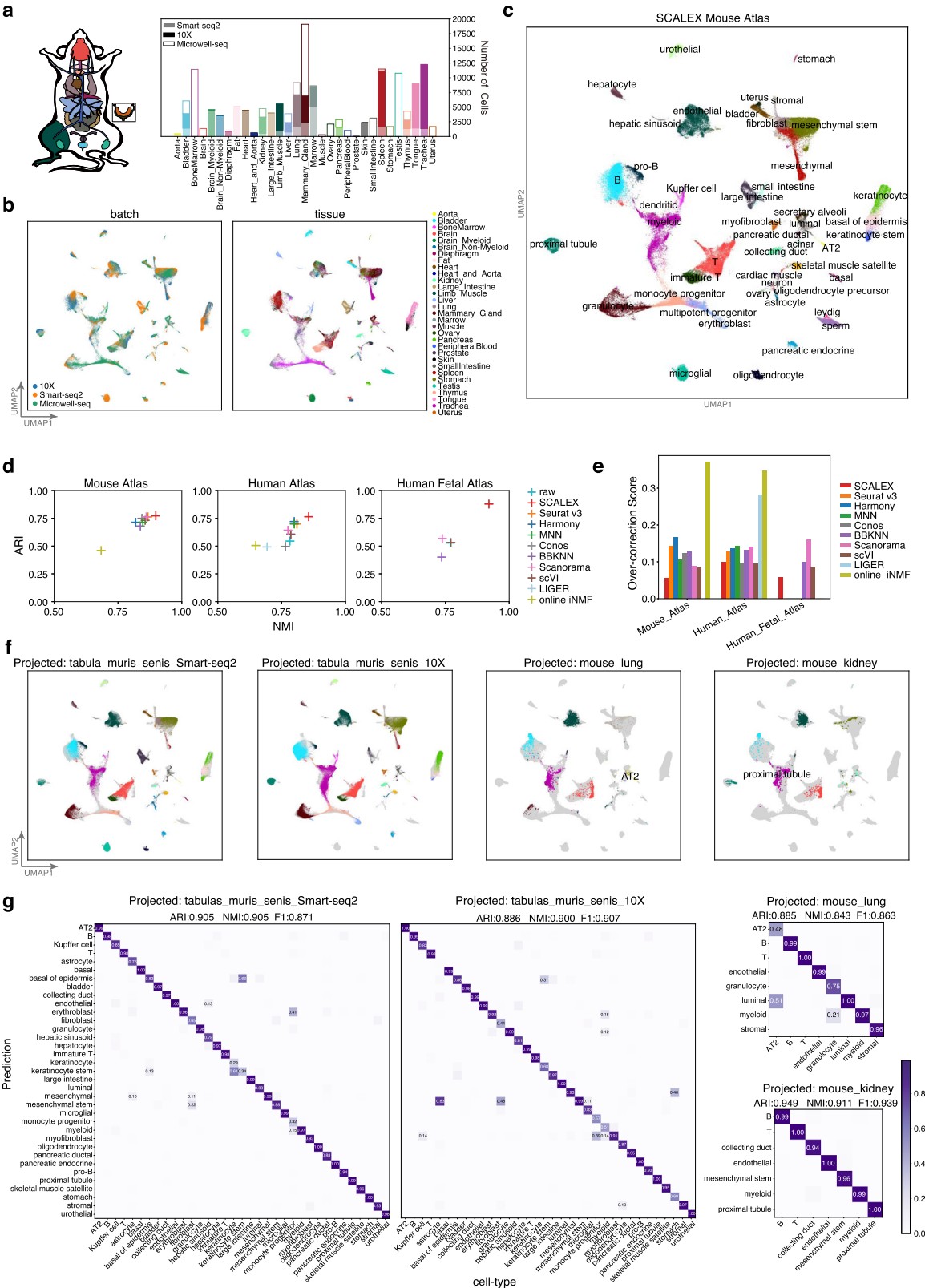

**Fig. 4 | Construction of an expandable mouse single-cell atlas. a** Datasets acquired using different technologies (Smart-seq2, 10X, and Microwell-seq) covering various tissues used for construction of the mouse atlas. **b** UMAP embeddings of the Mouse Atlas dataset colored by batch and tissue. **c** UMAP embeddings of the Mouse Atlas after SCALEX integration, colored by cell-type. **d** Scatter plot showing a quantitative comparison of the ARI score (y-axis) and the NMI score (x-axis) based on the Leiden clustering results on the latent space based on the Human Atlas, Mouse Atlas, and Human Fetal Atlas datasets. **e** Comparison of over-correction score by the indicated methods based on the Human Atlas, Mouse Atlas, and Human Fetal Atlas datasets. **f** UMAP embeddings of the common cell space obtained by using SCALEX to project the two *Tabula Muris Senis* data batches and two mouse tissues (lung and kidney) data batches onto the Mouse Atlas dataset. Cells are colored by cell-type with light gray shadows representing the original Mouse Atlas dataset. **g** Confusion matrix of the cell-type annotations by SCALEX and those in the original studies. Color bar represents the percentage of cells in confusion matrix $C_{ij}$ known to be cell-type $i$ and predicted to be cell-type $j$.

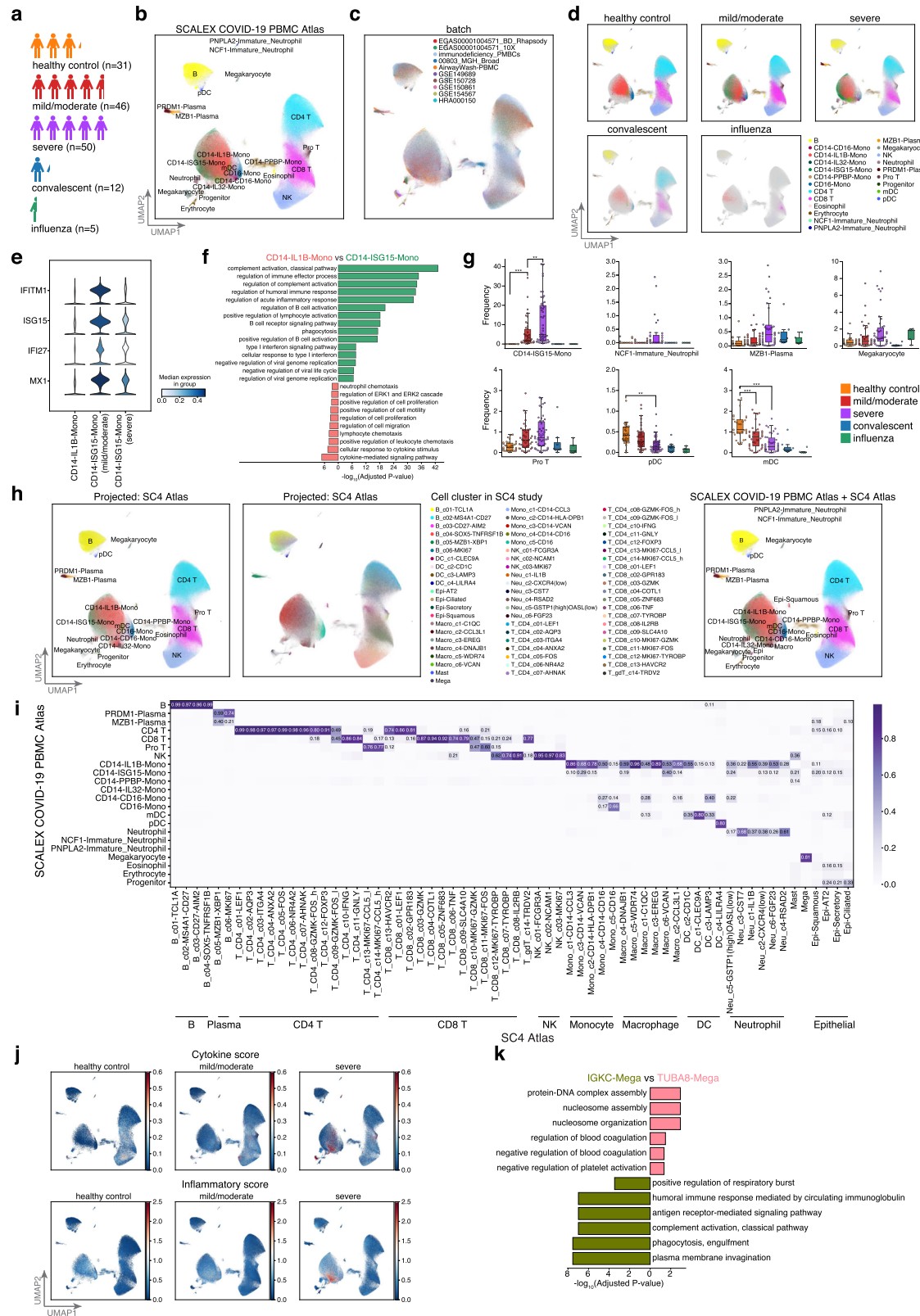

individual studies that served as the basis for our SCALEX COVID-19 PBMC atlas.

**Online integration of the SCALEX COVID-19 PBMC Atlas with the SC4 consortium study**

Our analysis based on the SCALEX COVID-19 PBMC Atlas yielded findings consistent with two conclusions from the Single Cell Consortium for COVID-19 in China (SC4) study, a recent large-scale effort that generated a single-cell atlas of over 1 million cells from 171 COVID-19 patients and 25 healthy controls[7] (Supplementary Fig. 18a). First, both studies observed the same set of immune cell subpopulations which displayed differential associations with COVID-19 severity. The proportions of CD14 monocytes, megakaryocytes, plasma cells, and pro T cells were elevated with increasing disease severity, while the

**Fig. 5 | Online integration of COVID-19 PBMC Atlas. a** The COVID-19 PBMC Atlas dataset composition, including healthy controls and influenza patients, as well as mild/moderate, severe, and convalescent COVID-19 patients. **b, c** UMAP embeddings of the COVID-19 PBMC Atlas after SCALEX integration colored by cell-type (**b**), and by batch (**c**). Note that here in order to keep the same UMAP embedding space as (**h**), we show the UMAP embeddings of the SCALEX COVID-19 PBMC Atlas after projecting SC4 data, The UMAP embedding of SCALEX COVID-19 PBMC Atlas alone is shown in Supplementary Fig. 16. **d** UMAP embeddings of the COVID-19 PBMC Atlas separated by disease state. Cells are colored by cell-type with light gray shadows representing the other disease state cells. **e** Stacked violin plot of differentially-expressed ISGs among CD14 monocytes across disease states. **f** GO terms enriched in the differentially-expressed genes for CD14-IL1B-Mono and CD14-ISG15-Mono cells. Hypergeometric test, $p$-values were adjusted using the Benjamini-Hochberg method. **g** Cell-type frequency across healthy ($n = 31$) and influenza controls ($n = 5$), and among mild/moderate ($n = 46$), severe ($n = 50$), and convalescent ($n = 12$) COVID-19 patients. Dirichlet-multinomial regression was used for pairwise comparisons, two-sided $t$-test, CD14-ISG15-Mono, healthy control vs mild/moderate: $p=9.78\times10^{-10}$, mild/moderate vs severe: $p=0.0057$; pDC, healthy control vs severe: $p = 0.0053$; mDC, healthy control vs mild/moderate: $p = 0.00072$, healthy control vs severe: $p = 1.05 \times 10^{-5}$. \*\*\*$p < 0.001$, \*\*$p < 0.01$, \*$p < 0.05$. Midline, median; boxes, interquartile range; whiskers, 1.5× interquartile range. **h** UMAP embeddings of the common cell space obtained by using SCALEX to project the SC4 Atlas (Single Cell Consortium for COVID-19 in China). Cells are colored by cell-type from label transfer based-on the locations in the COVID-19 PBMC Atlas dataset (left) and cell clusters in original SC4 study (middle), and Unified UMAP embeddings combining the SCALEX COVID-19 PBMC Atlas and the SC4 Atlas (right). **i** Confusion matrix of the cell-type annotations by SCALEX and those in the original studies. Color bar represents the percentage of cells in confusion matrix $C_{ij}$ known to be cell-type $i$ and predicted to be cell-type $j$. **j** UMAP embeddings of the SCALEX COVID-19 PBMC Atlas colored by the cytokine score and the inflammatory score. **k** GO terms enriched in the differentially-expressed genes for TUBA8-Mega and IGKC-Mega cells. Hypergeometric test, $p$ values were adjusted using the Benjamini-Hochberg method.

proportion of pDC and mDC cells decreased (Fig. 5g). Second, based on calculating the same cytokine score and inflammatory score (defined in the SC4 study) for the cells in our SCALEX COVID-19 PBMC Atlas, we confirmed that the monocyte subpopulations are associated with cytokine storms triggered by SARS-Cov2 infection and are further elevated in severe patients[77] (Fig. 5j, "Inflammatory and cytokine score analysis" in Methods, adjusted $p$-value < 0.01).

SCALEX's online integration capacity enables us to project the SC4 consortium dataset into the cell space of the SCALEX COVID-19 PBMC Atlas. We found that the cell-types of two atlases were well-aligned (Fig. 5h, i, Supplementary Fig. 18b, c). Integration of the SC4 data further substantially improved both the scope and resolution of the SCALEX COVID-19 PBMC Atlas. First, this data added macrophages and epithelial cells to the cell space, enabling investigation of their potential involvement in COVID-19. The integration also supported more precise characterization of specific cell subpopulations. For example, the megakaryocyte population, not distinguished in either the SCALEX COVID-19 PBMC Atlas or the SC4 Atlas (Supplementary Fig. 16c), were divided into two subpopulations in the combined atlas after projection of SC4 (Fig. 5h). An exploratory functional analysis of the differentially expressed genes in these two newly delineated megakaryocyte subpopulations (TUBA8-Mega and IGKC-Mega, Supplementary Fig. 18d, e) revealed enrichment for the GO terms "humoral immune response" for IGKC-Mega cells, yet enrichment for "negative regulation of platelet activation" for TUBA8-Mega cells (Fig. 5k). These results illustrate how the continuously expandable single-cell atlases generated using SCALEX capitalize on existing large-scale data resources and also facilitate the discovery of new biological and biomedical insights.

## Discussion

Single cell studies are becoming more and more prevalent, growing larger and larger in scale, and expanding in the scope of sample types, often with quite heterogenous cell subsets. Thus, there is a great need for data integration tools to accurately and efficiently handle these Atlas-level datasets[33]. Further, there is also a need for online integration capacity to continuously incorporate incoming new data with existing integrations without having to recalculate from scratch[24]. By design, SCALEX learns a generalized projection function to project heterogeneous single-cell data into a common cell-embedding space, enabling it to achieve bona fide online data integration. SCALEX is also computationally efficient, and preserves biological variations and avoids over-correction when integrating partially overlapping datasets.

These features make SCALEX particularly useful for Atlas-level datasets, allowing the integration of many single-cell studies to support ongoing, very large-scale research programs throughout the life sciences and biomedicine. We speculate that use of SCALEX to project single-cell datasets from highly diverse cancer types to construct a pan-cancer single-cell atlas may lead to the discovery of previously unknown cell-types that are common to divergent carcinomas and that function in pathogenesis, malignant progression, and/or metastasis.

## Methods

### Overview of the SCALEX model

SCALEX applies a variational autoencoder (VAE) to project the different batches of datasets into the same batch-invariant low-dimensional embeddings by learning a batch-free encoder and a batch-specific decoder simultaneously. Since the encoder and decoder are coupled to learn a batch-free encoder, a batch label ($b$) is only exposed to the decoder within the domain-specific batch normalization (DSBN), thus the decoder captures the batch information while the encoder learns the domain-invariant features. In the encoder, SCALEX takes the input expression profile (**x**) across all the batches as a whole mixture distribution without distinguishing their batch sources and extracts their mean (**μ**) and variance (**σ**²) of the latent representations (**z**) in a 10-dimension embedding space to learn their global data structure. A standard multivariate Gaussian prior is used for **z**, while the approximated distribution of **z** is re-parameterized by $\mathbf{z} = \boldsymbol{\mu} + \boldsymbol{\sigma} * \boldsymbol{\varepsilon}$, where $\boldsymbol{\varepsilon}$ is sampled from $\mathbb{N}(0, \mathbf{I})$. In the decoder, SCALEX maps the latent representations with batch label ($b$) back to their original profile. To enable the decoder to capture the batch-specific variations, a DSBN layer is applied to learn a batch-specific normalization for each batch label ($b$), before transforming them back to their original profile with the new batch variations. To learn the global distribution to avoid overcorrection on partially overlapping datasets, within each mini-batch in the training process, SCALEX randomly samples data from all batches and trains on them together with Batch Normalization to smooth the batch-specific shifts and align to the global distribution. Once trained, the encoder of SCALEX is generalized to any batches and serves as a universal function for globally mapping different batches of datasets into the same batch-invariant space.

Training SCALEX is to maximize the log-likelihood of the observed single-cell sequencing data (**x**):

$$logp\,(\mathbf{x}) = \log \int_{\mathbf{z}} p(\mathbf{x}, \mathbf{z})d\mathbf{z} \qquad (1)$$

$$\geq E_{q(\mathbf{z}|\mathbf{x})}\left[log\frac{p(\mathbf{x}, \mathbf{z})}{q(\mathbf{z}|\mathbf{x})}\right] \qquad (2)$$

$$= \mathscr{L}_{ELBO}(\mathbf{x}) \qquad (3)$$

Then the loss function is transformed into the evidence lower bound (ELBO). While the ELBO can be further decomposed into two terms:

$$\mathcal{L}_{ELBO}(\mathbf{x}) = E_{q(\mathbf{z},|\mathbf{x})}\left[log\, p(\mathbf{x}|\mathbf{z})\right] - D_{KL}(q(\mathbf{z}|\mathbf{x}) \parallel p(\mathbf{z})) \quad (4)$$

The first term is the reconstruction term, which minimizes the distance between the generated output data ($\mathbf{x}'$) and the original input data ($\mathbf{x}$), calculated as the binary cross entropy between $\mathbf{x}'$ and $\mathbf{x}$. The second term is the regularization term, which minimizes the Kullback-Leibeler divergence between posterior distribution $\mathbb{N}(\boldsymbol{\mu}, \boldsymbol{\sigma}^2)$ and prior distribution $\mathbb{N}(0, \mathbf{I})$ of latent representations ($\mathbf{z}$). To enable a more flexible alignment under the latent space, we adjusted the coefficient of the second term to 0.5 after hyper-parameter optimization via a grid search; thus, the final loss function is:

$$\mathcal{L}_{ELBO}(\mathbf{x}) = E_{q(\mathbf{z}|\mathbf{x})}\left[log\, p(\mathbf{x}|\mathbf{z})\right] - 0.5*D_{KL}(q(\mathbf{z}|\mathbf{x}) \parallel p(\mathbf{z})) \quad (5)$$

The overall network architecture of SCALEX consists of an encoder and a decoder. The encoder is a two-layer neural network (fully connected [1024]-BN-ReLU-fully connected [10]) for mean ($\boldsymbol{\mu}$) and variance ($\boldsymbol{\sigma}^2$) of the 10-dimension latent representations ($\mathbf{z}$) using a reparameterization to obtain latent representations ($\mathbf{z}$), and the decoder has only one layer (no hidden layer), directly connecting latent representations ($\mathbf{z}$) to the output ($\mathbf{x}'$) (fully connected-DSBN-Sigmoid) with domain-specific batch normalization, where the latent representations ($\mathbf{z}$) and batch label ($b$) are provided as input, and a Sigmoid activation function. We used the Adam[78] optimizer with a 5e-4 weight decay and betas (0.9, 0.999, the exponential decay rate for the first and second moment parameters) to optimize the model under the learning rate 2e-4. We adopted mini-batch strategy to iteratively optimize the model, in each mini-batch, we randomly sampled data from all batches instead of from the same batch, and the mini-batch size for training input is 64. The maximum number of training iterations is 30,000 and an early stopping is triggered when there has been no improvement for 10 epochs. The hyper-parameters are chosen after a grid search. SCALEX is very robust with all of these hyper-parameters, all of the results in this manuscript are produced under the same parameters.

## Domain-specific batch normalization (DSBN)
Batch normalization (BN)[32] is a widely used training technique in deep neural networks to reduce internal covariate shifting. A BN layer whitens activations within a mini-batch of samples followed by scaling and shifting with learned affine parameters $\gamma$ and $\beta$. For a mini-batch of samples: $\mathcal{B} = \{x_{1\ldots m}\}$;

$$\mu_{\mathcal{B}} = \frac{1}{m}\sum_{i=1}^{m} x_i \quad (6)$$

$$\sigma_{\mathcal{B}}^2 = \frac{1}{m}\sum_{i=1}^{m} (x_i - \mu_{\mathcal{B}})^2 \quad (7)$$

$$\hat{x}_i = \frac{x_i - \mu_{\mathcal{B}}}{\sqrt{\sigma_{\mathcal{B}}^2 + \epsilon}} \quad (8)$$

$$y_i = \gamma\hat{x}_i + \beta \equiv BN_{\gamma,\beta}(x_i) \quad (9)$$

Where $\mu_{\mathcal{B}}$ is the mini-batch mean, $\sigma_{\mathcal{B}}^2$ is the mini-batch variance, $\hat{x}_i$ is the normalized output by $\mu_{\mathcal{B}}$ and $\sigma_{\mathcal{B}}^2$, $y_i$ is the BN output by scaling and shifting $\hat{x}_i$ with parameters $\gamma$ and $\beta$, and $\epsilon$ is a constant added to the mini-batch variance for numerical stability.

Domain specific batch normalization (DSBN)[31] is a combination of multiple sets of BN specific to each domain. DSBN learns domain-specific affine parameters $\gamma_d$ and $\beta_d$ for each domain, $d$ is the domain label; here, domain represents different batches. In the neural network, DSBN serves like multi-channel BN and switches to the corresponding BN given the domain label $d$. The DSBN layer can be written as:

$$y_d = \gamma_d\hat{x}_d + \beta_d \equiv DSBN_{\gamma_d,\beta_d}(x_d, d) \quad (10)$$

where $d$ is the batch label, and $\gamma_d$ and $\beta_d$ are domain-specific affine parameters for domain $d$.

DSBN could capture the domain-specific information by estimating mini-batch statistics by learning affine parameters for each domain separately, thus enabling the network to learn the domain-invariant features.

## Preprocessing for scRNA-seq
We downloaded gene expression matrices and preprocessed them using the following procedure: i). Cells with fewer than 600 genes and genes present in fewer than 3 cells were filtered out. ii). Total counts of each cell were normalized to 10,000. iii). Values of each gene were subjected to log transformation with an offset of 1. iv). The top 2000 highly variable genes were identified. v). Values of each gene were normalized to the range of 0-1 within each batch by the *MaxAbsScaler* function in the *scikit-learn* package in Python. The processed matrix was used as input for the SCALEX model for downstream differential gene expression analysis.

For the *human fetal atlas* dataset, we collected two batches (batch GSE156793, which contains 4,062,980 cells by sciRNA-seq3, and batch GSE134355, which contains 254,266 cells by Microwell-seq). We then selected the cells from the common tissues (1,369,619 cells) for integration and computational efficiency benchmarking (down-sampled from different data sizes including 10 K, 50 K, 250 K, 1 M, and 4 M).

## Preprocessing for scATAC-seq
We downloaded open chromatin profile matrices (peaks or bins), merged them by peaks (or bins), and processed them using the following procedure: i). The combined matrix was binarized and filter bins with fewer than 3 cells. ii). The top 30,000 most variable peaks (or bins) were selected using the *select_var_feature* function in the *EpiScanpy*[79] package. iii). Total counts of each cell were normalized to the median of the total counts of all cells by using the *normalize_total* function, with parameters *target_sum*="None" in the *Scanpy*[80] package. iv). Values of each peak (or bin) were normalized to the range of 0-1 within each batch by the *MaxAbsScaler* function in the *scikit-learn* package in Python. The processed matrix was used as input for the SCALEX model.

## Preprocessing for cross-modality data (scRNA-seq and scATAC-seq)
We first created a gene activity matrix by the *GeneActivity* function in the *Signac*[81] R package to quantify the activity of each gene from scATAC-seq data. We then combined gene activity score matrix with scRNA-seq data matrix as two individual "batches" for integration. The subsequent preprocessing followed the same preprocessing used for the scRNA-seq data (above).

## Clustering
For Harmony, MNN, Conos, BBKNN, Scanorama, scVI, LIGER, and online iNMF, we used their latent features with method specific default dimensions for further clustering. For Seurat v3, we initially performed integration and obtained the 2000-dimensional latent feature vectors following the standard workflow, and then we used PCA for

dimensionality reduction because 2000-dimensional latent feature vectors are too high to directly cluster. Finally, we used 50-dimensional PCA latent feature vectors for clustering. For Conos and BBKNN, since they do not provide latent feature vectors after integration (and we failed to extract the latent feature vectors from their constructed either neighborhood or joint graphs), we used UMAP features for downstream clustering.

To ensure a fair comparison, we used *scanpy.tl.leiden* and *scanpy.tl.louvain* functions for clustering with *resolution*=0.5. For BBKNN and Conos, since *resolution*=0.5 generates too many clusters, we also included clustering results of with *resolution* = 0.05, which were used in our benchmark comparison (more details in Supplementary Dataset 2).

### Visualization
UMAP algorithm[46] was used for visualization. We applied the *neighbors* function from the Python package *Scanpy* with the parameters *n_neighbors*=30 and *metric*="Euclidean" for computing the neighbor graph, followed by *umap* function with *min_dist*=0.1 to visualize cells in a two-dimensional space. Tissue anatomy diagrams are generated by *gganatogram* (v2) R package[82,83].

### Adjusted Rand Index
The Rand Index (RI) computes a similarity score between two clustering assignments by considering matched and unmatched assignment pairs, independent of the number of clusters. The Adjusted Rand Index (ARI) score is calculated by "adjust for chance" with RI as follows:

$$ARI = \frac{RI - Expected\_RI}{\max(RI) - Expected\_RI} \tag{11}$$

If given the contingency table, then ARI can also be represented by:

$$ARI = \frac{\sum_{ij}\binom{n_{ij}}{2} - \frac{\left[\sum_i \binom{a_i}{2}\sum_j \binom{b_j}{2}\right]}{\binom{n}{2}}}{\frac{1}{2}\left[\sum_i \binom{a_i}{2} + \sum_j \binom{b_j}{2}\right] - \frac{\left[\sum_i \binom{a_i}{2}\sum_j \binom{b_j}{2}\right]}{\binom{n}{2}}} \tag{12}$$

The ARI score is 0 for random prediction and 1 for perfectly matching.

### Normalized mutual information

$$NMI = \frac{I(P;T)}{\sqrt{H(P)H(T)}} \tag{13}$$

Where P and T are categorical distributions for the predicted and real clustering, I is the mutual entropy, and H is the Shannon entropy.

### Silhouette score
We used the silhouette score to assess the separation of biological populations with the function *silhouette_score* in the *scikit-learn* package in Python. The silhouette score was computed by combining the average intra-cluster distance (a) and the average nearest-cluster (b) for each cell.

$$silhouette\ score = \frac{b - a}{\max(a, b)} \tag{14}$$

Here, we took UMAP embeddings as input to calculate silhouette score.

### Batch entropy mixing score
Batch entropy mixing score (adapted from "entropy of batch mixing"[19]) was used to access the regional mixing of cells from different batches, with a high score suggesting that cells from different batches are well mixed together.

The batch entropy mixing score was computed as follows:
(1) Calculated the proportion $Pi$ of cell numbers in each batch to the total cell numbers.
(2) Randomly chose 30 cells from all batches.
(3) Calculated the 30 nearest neighbors for each randomly chosen cell.
(4) The regional mixing entropies for each cell were defined as:

$$pi' = \frac{\frac{pi}{Pi}}{\sum_{i=1}^{n}\frac{pi}{Pi}} \tag{15}$$

$$E = \sum_{i=0}^{n} pi'\log(pi') \tag{16}$$

where $pi$ is the proportion of cells from batch $i$ in a given region, such that $\sum_{i=0}^{n} pi = 1$, $pi'$ is a correction item to eliminate the deviation caused by the different cell numbers in different batches. The total mixing entropy was then calculated as the sum of the regional mixing entropies.
(5) Repeated (2)-(4) for 10 iterations with different randomly chosen cells and calculated the average, E, as the final batch entropy mixing score.

Note that to mitigate the effect of misalignment of batch-specific cell-types, we calculated the batch entropy mixing score only based on cells from cell-types that are common in different batches.

### Local inverse Simpsons Index (LISI)
The LISI metric was proposed by Korsunsky et al. 2019[20] to assess batch and cell-type mixing. We calculated integration LISI (iLISI) and cell-type LISI (cLISI) values using the *compute_lisi* function in the *lisi* R package. UMAP embeddings, batch labels, and cell-type labels were used as input in calculation. Briefly,

$$LISI(x_i) = \frac{1}{\sum_{y \in Y} P(y,|x_i)^2} \tag{17}$$

where $x_i \in \{x_1, x_2, \ldots, x_N\}$ is the $i$-th cell's UMAP embeddings in the dataset of size $N$, and $Y$ is the set of unique values with respect to the type of LISI we are computing (i.e., $Y$ is the values of "batch label" for calculating iLISI and the value of "cell-type label" for calculating cLISI). The probability $P(y,|x_i)$ refers to the "relative abundance" of the covariate $y$ within KNN (k-nearest neighborhood) of $x_i$. A Gaussian kernel-based distribution of neighborhoods was used and the perplexity was fixed to 30.

### Over-correction score
We defined an over-correction score to assess the level of over-correction problem, based on calculating the percentage of cells with inconsistent cell-types in each cell's neighborhood. We calculated the over-correction score over all cells, and for each cell i we averaged the frequency of the k-nearest neighboring cells with distinct cell-types to the cell i (see the following equation).

$$over\_correction\ score = 1 - \frac{1}{n*k}\sum_{i=1}^{n}\sum_{j=1}^{k} I\left(cell\_type_i, cell\_type_j\right) \tag{18}$$

where n is the total cell number, k represents the k-nearest neighbors of each cell, the cell-type of the cell i is $celltype_i$, the cell-type of the neighboring cell j is $celltype_j$, and $I$ is an indicative function defined as:

$$I\left(cell\_type_i, cell\_type_j\right) = \begin{cases} 1 \text{ if } cell\_type_i = cell\_type_j \\ 0 \text{ if } cell\_type_i \neq cell\_type_j \end{cases} \quad (19)$$

Formally, the over-correction score is a negative index, i.e., the higher the over-correction score, the more severe the extent of inaccurate mixing of cell-types.

**F1 score**

We calculate the F1 score by the function *f1_score* with *average*= "macro" in the *scikit-learn* package in Python.

**Single-cell integration benchmarking (scIB)**

The scores for all 12 examined metrics were calculated using the Python package *scIB*[50] with default parameters. The batch_correction_mean, bio_conservation_mean, and overall scores (rectangles) were calculated as described in the work of Luecken et al.[50] to assess the performances of different methods in terms of the batch removal, the conservation of biological variance, and the overall accuracy scores, respectively.

**Comparison with other integration methods**

We compared SCALEX to nine other batch effect removal methods (see below for specific details of each method). For each dataset as input for all methods, we performed the same filtration, followed by method-specific normalization, batch correction and visualization. Note that for visual comparison, we also included the embeddings of the raw input data, wherein we performed dimensionality reduction by Principal Component Analysis (PCA)[84] followed by UMAP visualization to see the batch effects. No correction function was used. All parameters were kept as default values.

Scanorama (v1.6). We performed the preprocessing pipelines as stated above (as the same below), and used the *Scanpy* and *scanorama* Python packages for integration. For the *highly_variable_genes* function, we set *flavor*="seurat", *batch_key*="batch", and *n_top_genes*=2,000. After extracting highly variable genes, we divided the datasets according to the batch labels and formed a new list of datasets as the input for the *correct_scanpy* function. The integration matrix was kept for downstream analysis. All other parameters were kept their default values.

BBKNN (v1.3.12). We used *Scanpy* and *bbknn* Python packages and followed the suggested pipelines for integration. For the *highly_variable_genes* function, we set *flavor*="seurat", *batch_key*="batch", and *n_top_genes*=2,000. After selecting cell neighbors at the low-dimensional space from the PCA analysis, we performed the *bbknn* function with *neighbors_within_batch*=5, *n_pcs*=20, and *trim*=0. All other parameters were default.

scVI (scvi-tools, v0.11): We used the *scvi* Python package and followed the suggested pipelines. Batch information was added to the VAE model by setting n_batch.

Seurat v3 (v3.2.3): We used the *Seurat* R package and followed the standard integration workflow. We normalized different batches of a dataset separately. For the *FindVariableFeatures* function, we set *selection.method*="vst" and *nfeatures*=2000 to select 2000 highly variable genes for each batch of a dataset. For the *FindIntegrationAnchors* function, we set *k.filter*=100. All other parameters were kept at default values. If the number of input cells in a dataset exceeded 50,000, we employed the reciprocal PCA and reference-based integration to improve computational efficiency.

Harmony (v1.0): We used the *harmony* R package. We created a *Seurat* object with all cells and performed the standard workflow. After PCA, we used the *RunHarmony* function for integration. All parameters were default.

Conos (v1.3.1): We used the *Conos* R package. For each batch of dataset, we used the *basicSeuratProc* and *RunTSNE* functions for pre-processing. After that, we built a joint graph using the *buildGraph* function with *k*=30 and *k.self*=5. All other parameters were default.

MNN (FastMNN, v0.3.0): We used the *SeuratWrappers* R package. We created a Seurat object with all cells and performed the standard workflow. Then we used the *RunFastMNN* function with default parameters for integration.

Online iNMF and LIGER (LIGER, v1.0.0): We used the *rliger* R package. For the online iNMF method, we used the online_iNMF function with *k*=20, *miniBatch_size*=5,000 and *max.epochs*=5. For the LIGER method, we used the *optimizeALS* function with *k*=20. All other parameters were the default values. Different from other methods, online iNMF only loads one mini-batch from the whole data in the HDF5 file format (converted from the original data format by the *rhdf5* R package) for a memory-efficient implementation; accordingly, a file conversion issue with the down-sampled *human fetal atlas* dataset of 4 M data size prevented online iNMF from calculating computational efficiency with the 4 M.

scJoint: We used the *scJoint* Python package. We pre-processed the data into the standard input format for *scJoint*, and then modified the *config.py* file in the *scJoint* package and set the same training config parameters as used in the tutorial of "Analysis of PBMC data from 10x Genomics using scJoint" (https://github.com/sydneybiox/scJoint/blob/main/tutorial/Analysis%20of%2010xGenomics%20data%20using%20scJoint.ipynb).

bindSC (v1.0.0): We used the *bindSC* R package. Following the tutorial, we first performed dimension reductions for gene expression, for the gene activity scores, and for the chromatin accessibility profiles, using the *dimReduce* function with *K*=30. Subsequently, we ran the *BiCCA* function with *lambda*=0.5, *alpha*=0.5, and *K*=20. All other parameters were default.

**Cell-type annotation by clustering**

This type of annotation was used for de novo annotation of a single-cell dataset. We used a Leiden clustering[85] method for cell clustering (specifically employing the *leiden* function from the Python package *Scanpy* with default parameters). Then for each cluster, we annotate its cell-type based on: i) cell-type annotations of each cell in the original study, if available, or ii) expression levels of canonical marker genes in each cell. A majority vote strategy was used when needed. Similar to Ren et al. 2021, we also employed a hierarchical annotation strategy, i.e., we first clustered all cells in a dataset into several major clusters, then for some big clusters, we further clustered them into minor clusters respectively.

**Single cell projection**

We defined single cell projection as the operation to convert high-dimensional single-cell data (e.g., gene expression profiles in scRNA-seq or open chromatin profiles in scATAC-seq) to low-dimensional representations in the common SCALEX cell-embedding space using the trained encoder.

**Cell-type annotation by label transfer**

This type of annotation was used for annotation of a new single-cell data batch using the annotations in a large single-cell dataset as a reference, or for *post hoc* annotations of unknown cell population(s) in a large dataset using new batches of data of known cell-types. Both scenarios require "single cell projection" (see details below).

The basic idea of cell-type annotation by label transfer is based-on that the same cell-types will occupy the same locations in the low-dimensional SCALEX cell-embedding space, thus cell-type annotation in one data batch can be transferred to another data batch, for the cells

positioned at the same locations. Technically, we used the *KNeighborsClassifier* function from the *scikit-learn* package to train a prediction model, using the representations (in the low-dimensional cell-embedding space) of the single-cell data with known cell-type labels as input. We then used this model to make cell-type predictions for cells without annotations using their representations (in the low-dimensional cell-embedding space) as input. For comparison, label transfer for online iNMF follows the same procedures as SCALEX by predicting the cell-type based-on the projected locations.

### Similarity matrix and confusion matrix
We used similarity matrix to evaluate the congruence of two different batches for the same cell-types in the common cell-embedding space. Technically, we merged all cells with the same cell-type label and calculated an average representation (in the low-dimensional cell-embedding space) for the cell-type. This was repeated for all cell-types. We then calculated the similarity matrix $S = [S_{ij}]$ for the cell-type similarities of the two batches, where $S_{ij}$ is the Pearson correlation coefficient between the average representation of cell-type $i$ in data_batch_1 and the average representation of cell-type $j$ in data_batch_2.

We used the confusion matrix to evaluate the accuracy of cell-type annotations (prediction) when a gold-standard annotation is available, which is typical for "cell-type annotation by label transfer" (see above). In cell-type annotation by label transfer, we predict the cell-types for a single-cell data_batch_1, using the annotations in another data_batch_2. When data_batch_1 was already annotated with cell-types, we can calculate the confusion matrix $C=[C_{ij}]$ to compare the cell-type predictions with the existing cell-type annotations, where $C_{ij}$ equals the percentage of cells known to be in cell-type $i$ and predicted to be in cell-type $j$.

### Generation of partially overlapping datasets
To simulate partially overlapping datasets from the *pancreas* dataset, we used the pancreas_celseq2 and pancreas_smartseq2 data batches, and worked with only six cell-types (alpha, beta, ductal, acinar, delta, gamma). For each simulated partially overlapping dataset, we randomly selected three to six cell-types from each batch, and counted the number of the common cell-types, which was used as the indicator for the overlapping level (whole integers, 0 to 6). We required the union of cell-types in the newly simulated partially overlapping dataset to cover all six cell-types.

For the *PBMC* dataset, we used both of the two data batches and worked with twelve cell-types (B, CD4 T, CD4 naive T, CD8 T, CD8 naive T, DC, HSC, Megakaryocyte, NK, monocyte-CD14, monocyte-FCGR3A, pDC). We used the same down-sampling strategy as for the *pancreas* dataset (above).

### Analysis of changes in cell-type frequency across multiple conditions
To identify differences in cell-type frequency among the scRNA-seq data from the mild/moderate, severe, convalescent COVID-19 patients, as well as the healthy and influenza patient controls, we applied a Dirichlet-multinomial regression model. This model accounts for the constraint that the cell frequencies in a scRNA-seq data are not independent of each other. In detail, we normalized the regression coefficients to a standard normal distribution and calculated a z score, and then conducted significance testing based on the regression model generated by the *DirichReg* function in the R package *DirichletReg* (v0.7).

### Differential gene expression analysis and Gene Ontology term enrichment analysis
Differential gene expression analysis was performed on all expressed genes using the *rank_genes_groups* function with *method*="*t*-test" in the *Scanpy* package, for two certain cell-types in a COVID-19 single-cell atlas. A gene was considered differentially expressed when a log2-fold change was >1 in the two conditions in comparison, and the Benjamini-Hochberg adjusted *p*-value was < 0.01. The top 200 highly expressed genes sorted by scores (implemented in *Scanpy*) of each cell-type were used as the input for GO analysis, and enriched GO terms were acquired for each group of cells of the "GO_Biological_Process_2018" dataset using the Python package *gseapy* (v0.10.1).

### Inflammatory and cytokine score analysis
We defined the inflammatory score and the cytokine score for each cell following Ren et al. 2021[7], based on the expression of a defined collection of cytokine genes and inflammatory-response-related genes (Supplementary Dataset 3). We then calculated the cytokine and inflammatory scores from the raw gene expression profile using the *score_genes* function implemented in the *Scanpy*.

### Ablation studies using test-variants of SCALEX
To accomplish an accurate generalized encoder, the design of the full SCALEX included the following specific innovations:

1. an asymmetric autoencoder that inputs batch information only to the decoder (i.e., never to the encoder) (See diagram in Supplementary Fig. 1);
2. a DSBN layer in the decoder to release the encoder from the burden of capturing the batch-specific variations;
3. a mini-batching strategy that samples data from all batches simultaneously (rather than single batches iteratively) and thus more tightly follows the same overall distribution of the full input dataset; this strategy includes a Batch Normalization layer in the encoder that adjusts the deviation of each mini-batch and aligns them to the overall input distribution.

We conducted ablation studies to investigate the contributions of each design element of SCALEX. That is, we analyzed the performance (for integration and projection tasks) of the full SCALEX and four SCALEX "test-variants", each with a distinct network architecture (Supplementary Figs. 19–21). These can be summarized as follows:

### Full SCALEX (referred as "Baseline" in the following)
This Baseline model includes an encoder without batch labeling, sampling from all batches with Batch Normalization, decoder with DSBN, and beta=0.5. All other ablations are compared relative to this.

### Encoder with batch label
A test-variant with Baseline including an encoder with batch label as input. We found that the integration performance of this variant is similar with the full SCALEX, showing only a slight reduction in the evaluation scores (Supplementary Fig. 19). However, the real issue is this: this addition of batch information at the beginning precludes online integration of newly arriving data. Put another way, this SCALEX test-variant is not capable of integrating single-cell data in a truly online manner.

### Decoder without DSBN
A test-variant of Baseline removing the DSBN layer from decoder. The DSBN layer combines multiple batch-specific Batch Normalization layers to capture the batch-specific information; this approach has been demonstrated as effective for domain adaption, as it provides a weak alignment across different domains. We observed an obvious drop in the integration performance (in terms of all evaluation scores) and a slight drop in the projection performance of this SCALEX test-variant, based-on the UMAP embeddings (Supplementary Figs. 19, 20).

### Sampling by batch without the Batch Normalization (BN) layer in the encoder
Removing the Batch Normalization layer from the encoder, and each mini-batch is sampled by batch instead of from all the batches. The

integration performance of this test-variant was obviously worse than full SCALEX (Supplementary Fig. 19). The projection performance also dropped obviously, with clear deviations from the common cell-embedding space (Supplementary Fig. 20).

### Regular autoencoder

A test-variant that uses a regular autoencoder instead of a VAE framework. This variant performed the worst among all the variants we tested, for both the integration and projection tasks (Supplementary Figs. 19, 20).

Note that we also explored altering the beta factor. Replacing the beta factor as 1 instead of 0.5. The integration performance of test variant of SCALEX for beta factor of 1 is worse than the SCALEX for beta factor of 1 (Supplementary Fig. 21).

### Reporting summary

Further information on research design is available in the Nature Research Reporting Summary linked to this article.

## Data availability

All data analyzed in this study are publicly available; the data sources are detailed in Supplementary Dataset 1. All other relevant data supporting the key findings of this study are available within the article and its Supplementary Information files or from the corresponding author upon reasonable request.

## Code availability

SCALEX[86] is available at https://github.com/jsxlei/SCALEX. For reproducibility, the scripts for benchmarks and several case studies are also available in the above repository.

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

## Acknowledgements

We thank Jianbin Wang, Jin Gu and Fuchou Tang for helpful comments and advice. This work is supported by the State Key Research Development Program of China (Grant No. 2019YFA0110002, Q.C.Z.), the National Natural Science Foundation of China (Grants No. 32125007 and 91940306, Q.C.Z.), the Beijing Advanced Innovation Center for Structural Biology, and the Tsinghua-Peking Joint Center for Life Sciences. We thank the Tsinghua University Branch of China National Center for Protein Sciences (Beijing) for computational facility support. This work is also supported by the King Abdullah University of Science and Technology (KAUST) Office of Research Administration (ORA) under Award No. FCC/1/1976-44-01, FCC/1/1976-45-01, URF/1/4352-01-01, and URF/1/4663-01-01 (X.G.).

## Author contributions

Q.C.Z. conceived and supervised the project. L.X. designed and implemented the SCALEX model. L.X. and K.T. validated the SCALEX model. L.X., K.T., Y.L., W.N., and X.G. analyzed the results. L.X. and Q.C.Z. wrote the manuscript, with inputs from all the authors.

## Competing interests

The authors declare no competing interests.

## Additional information

**Peer review information** *Nature Communications* thanks Huachao Huang, Yang Xu, and the other, anonymous, reviewer(s) for their contribution to to the peer review of this work. Peer reviewer reports are available.

