## [Peer Review File · Nature Communications]

REVIEWER COMMENTS

Reviewer #1 (Expertise: scRNASeq, combo, DL):

In this manuscript, the authors have developed a deep-learning-based online integration method named SCALEX. The authors carried out a rigorous benchmark study with well-established methods in different datasets using different evaluation metrics. Their benchmark study demonstrated that SCALEX has overall better integration performance than other SOTA methods. They also showed that SCALEX has good generalizability and is capable of atlas integration. The unique advantage that SCALEX has over other integration methods is online learning, which can continuously build an integrated reference without retraining when new data arrive. Overall, this is a quality manuscript with substantial results to support its validity. However, some concerns and areas for improvement are discussed below.

Major:

1. The authors present a convincing argument that SCALEX has overall better generalizability than scVI and online iNMF. However, the major focus, “online learning” of SCALEX, is not as well supported and the manuscript needs more appropriate experiments to demonstrate the capabilities of online learning. New pancreas batches and new SKCM batches have a lot in common with the training datasets. Therefore, while projecting new batches into the existing space shows that SCALEX has good generalizability, this is not a challenging enough task to demonstrate that SCALEX is truly “online learning”. Based on my understanding of online learning, the arrival of new data should further fine tune the existing model (learning continues where the entire dataset is not available at any point during training). It is not clear in the manuscript if the authors did this for online iNMF. In the next section, the authors showed that SCALEX is capable of atlas data integration. These atlas datasets can be good examples to demonstrate that SCALEX is indeed an effective online learning method. The authors should perform the atlas data integration in an online manner. For example, the authors could start from one or two tissues across 3 atlas studies, and then fine tune the model with a new tissue. Gradually, the model should reach a similar outcome to the representation obtained by the full atlas data integration. It would be very useful to compare how this online integration result would be different from an atlas integration by SCALEX with all datasets present at one time.

2. In the label transferring task, the weighted F1 score reflects the overall accuracy of the model. However, it might not be a good choice to evaluate label transferring for single cell data. Since most single cell data are imbalanced, the macro F1 score can be a better way to evaluate how the model performs for non-dominant cell types. Indeed, the authors show >0.95 F1 scores in Fig. 3c, but the confusion matrices indicate the model is just mediocre for non-major cell types like “mast”, “quiescent stellate”, and “Schwann”. It is not clear whether SCALEX can accurately integrate single cell data that are very imbalanced in their cell type representations. This should either be tested further or discussed as a limitation of SCALEX.

3. Again, the good overlap between SC4 consortium and SCALEX COVID-19 PBMC atlas may only suggest good generalizability but not good online learning. Meanwhile, Pearson correlation is not a good metric to support the claim that “the cell-types of two atlases were well-aligned” (Fig. 5i). A confusion matrix after label transferring would better demonstrate the degree of alignment. As above, it seems SCALEX struggles to distinguish non-major DC as shown in Fig. 5i. This again may suggest a limitation that SCALEX is less able to distinguish non-major cell types.

4. Though cross-domain integration is not a focus of this study, the author brought up integrating RNA-seq and ATAC-seq data in Fig. 1f and g. The author should provide a more thorough and quantitative investigation on how SCALEX is competing against the many other existing cross-domain integration methods. Or, if this is not intended to be a major focus, this aspect should perhaps be removed from the study.

Minor:

1. The human heart data consists of single cells, nuclei and CD45+ enriched cells but the authors only consider addressing batch effects for nuclei data. It would be helpful to demonstrate how SCALEX handles batch effects that arise from how cells are collected.

2. Different integration methods may have different properties in terms of preserving data structure locally and globally. The authors did not provide details about how they clustered cells using integrated spaces learned by the different methods. Without these details, there is a concern that clustering parameters may have been chosen that artificially favor SCALEX leading to SCALEX having the highest ARI and NMI in every case. For example, Supplementary Fig 3 suggests that the author over-clustered cell types for other methods but selected a good clustering resolution for SCALEX. More details of how cells were clustered fairly could help clear up this potential appearance or existence of bias.

3. Some typos in the manuscript. For example, line 252: “model retraining” instead of “model retaining”.

Reviewer #2 (Expertise: pancreatic biology, scRNASeq, experimental):

The authors provide a method to integrate large scale datasets without requiring retraining and large processing times and computational power. This framework will be useful for curating consensus between different studies and useful for consorts where data is not generated in one go but rather generated parallels across labs and tissue types.

Specific questions to address:

1. Throughout the paper, the authors describe superiority of SCALEX in assessing cell types. The metrics used to determine overcorrection help to understand that to a certain extent. But the hindrance to biological interpretation is less clear to me. I miss the understanding of whether this is an issue of aesthetics in the UMAP projection or also the cluster assignment itself. For eg, a cell type assessed by Seurat can look split in the 2D embedding, but might have been assigned the same cluster number, thus still preserving the biological message. Changing clustering parameters to enable that is also a very important task for researchers. It would be useful to elaborate on this when describing the performance metrics.

2. In multimodality dataset integration, the authors show integration of scRNAseq and gene expression inferred by scATACseq. Although SCALEX shows a really good embedding of these two different assays, the comparison with existing specific pipelines (apart from supfig7) for such multiomics assays is lacking. How does SCALEX improve that? Plus, advances in multiomics assays now can enable scATAC and snRNA detection from same cell. Can SCALEX also improve analysis of such datasets?

3. The hepatocytes subsets are defined here by single gene differences. How biologically relevant are these subsets? How important is using SCALEX for this tissue type?

4. Based on fig2b, the overcorrection score looks to show that different techniques are performing differently based on the tissue type. So for example in case of heart data, there was hardly any difference in performance metrics between integration methods. Would you comment on that?

5. Figure 2D is a bit unclear. Extent of overlap as a concept needs to be described a bit more. Also, please attribute the exact subheadings in the methods section when referring to it in the main text. That way we are sure to be looking at the exact relevant subsection.

Why I find that this needs more explanation is because the abundance of mixing distinct cell types using other methods is quite a lot I feel. I do not recollect reading such discrepancies in existing literature on single cell atlas in pancreas.

6. On similar lines, intermediate polyhormonal cells in endocrine pancreas might be of interest. Is there a possibility that SCALEX, as it relies on existing knowledge of cell labels or markers, can miss out on them?

7. For figure 3E, this location of tumor cells, would it change with multiple heterogenous tumor samples? Is this distance between clusters that informative? Usually these distances can be very tricky to interpret. Could you comment on that?

8. For the novel cell type determined in pancreas atlas, is there any particular technique or sample from which these cells come from? Or are they present in multiple libraries? SLC16A7 or MCT2 expression has been investigated before in the context of islet function, but remained undetected (Zhao et al. (2001). Expression and Distribution of Lactate/Monocarboxylate Transporter Isoforms in Pancreatic Islets and the Exocrine Pancreas. Diabetes) What can be the reason that SCALEX identifies these cells? Can they be infiltrating cells? Again, does the guided labelling in the reference atlas impact the projection of such non-overlapping datasets?

9. COVID atlas looks very interesting! Have the authors used the PBMC atlas mentioned in earlier sections of the paper?

10. In figure 5g, where the frequencies are described, is it % of total cells in that sample? Without normalising for total number of cells, such fluctuating numbers can be tricky esp in diseased states.

11. So basically for atlases like covid and cancer, where knowledge of cell types and states is very recent, how will SCALEX perform in comparison to just looking into smaller subsets with more directed research questions? Could you comment on its strengths and drawbacks if any?

In conclusion, as my own expertise is closer to extracting biologically relevant points from these single cell investigations, I feel that the authors should emphasise the role of SCALEX in facilitating that. In terms of processing, I am convinced that it is very advantageous and shows promise. Discussing the advantages in biological consensus and translatability would be nice.

Reviewer #3 (Expertise: scRNASeq data analysis, inc. integration, whole-organism scale):

The authors have reported a single-cell data integration framework called SCALEX, which fundamentally uses a VAE, with a domain-specific batch normalization approach to align different single-cell RNA-seq datasets. They claim that their method does not require re-training on new datasets, and that as such their method can be deployed online with faster and more accurate performance than other existing methods. Although I appreciate the utility of being able to analyze large-scale single cell data online, I have doubts as to the extent of novelty of the method given the many other existing methods already in the space. In addition, I am not convinced that their method is more accurate than others out there sufficiently so to overlook the lack of novelty in the method itself. My specific major comments are below:

Major comments:

1. The SCALEX method is built upon the VAE framework. First, the authors themselves have previously reported this autoencoder for single-cell ATAC-seq data analysis: "We previously applied VAE and designed SCALE (Single-Cell ATAC-seq Analysis via Latent feature Extraction) to model and analyze single-cell ATAC-seq data". The majority of the novelty in using this type of framework for extracting latent features from single-cell data has already been reported in their previous publication. The only difference between the previous publication and the one being considered at present is the proposed domain-specific batch normalization (DSBN) in a batch-free encoder and batch-specific decoder, respectively. This is, in my opinion, not very novel as the specific normalization method is not new and only represents a minor change to SCALE. Furthermore, I have concerns as to the applicability of this method: as BN only shifts and scales data within a mini-batch of 64 cells (arguably in the context of atlas sized datasets it is not a representative sample size even), it presumably has limited ability to remove batch effects; it is unclear why the cell-embedding space of SCALEX should be batch-invariant, based on the method presented, as the theoretical model does not support such an assumption. Can the authors explain/give more intuition on that?

2. Most single-cell integration tools now demonstrate not only ability to integrate single-cell RNA-seq data, but also single-nuclei RNA-seq data. Due to the challenges associated with processing different sample and tissue types, single-cell atlas efforts now routinely use snRNA-seq to profile certain tissues/organs to avoid bias, while using scRNA-seq to profile other tissue types in the same atlas. Adipose and Neuronal tissues are prominent examples that call for snRNA-seq. Given that the authors claim to be superior for atlas-level datasets, and given that numerous other tools have been published in the same space, it would be very important for them to show applicability of their method to integration of not just scRNA/scRNA or snRNA/snRNA, but also scRNA/snRNA. The authors have integrated 'Harvard-Nuclei' and 'Sanger-Nuclei' data from the adult human heart atlas (<https://www.heartcellatlas.org/>). Besides these datasets, the atlas project also provided other two data sources: 'Sanger-Cells', 'Sanger-CD45'. If the authors can show integration of all of these from the same atlas, or at least align 'Sanger-Cells' with 'Sanger-Nuclei', it will demonstrate a more meaningful application of their method (i.e. building alignment between single-cell, single-nucleus data from a single atlas dataset).

3. The authors have carefully examined the design of SCALEX via an ablation study (Figs. S19, S20). Both BN and DSBN layers show clear impact on integration results. One major concern here is that the results from simple autoencoder are already not bad because the data preprocessing steps have the ability to adjust for batch effects. It appears possible that the data preprocessing already plays a significant role to remove the batch effects, and it is unclear how much of the correction is actually owing to the DSBN. It will be more attractive and convincing to users if the ablation study could be conducted based on more challenging scenarios.

Minor comments:

1. Somewhat relevant to point 3 above, most of the datasets selected by the authors (especially in the main figures) appear to be relatively less complex datasets, consisting mostly of tissues that harbor fewer and more distinct cell types, e.g. PBMCs, liver. However, the datasets that are more complex and involve multiple tissue types clearly do not show superior performance. For example, in Fig 4 the entire mouse atlas was integrated, including different technology types. While at first glance the results look ok, with closer examining we find cell types that do not belong together to be mixed, e.g. the neurons and cardiac muscle cells; some of the T cells are mixed with B cells; some cells unknown what type but all mixed together around the skeletal muscles and cardiac cells. In addition, some cells that belong together or nearby are no longer associated, including oligodendrocytes and their precursors, which are of the same lineage. The situation is more severe in S5 and S6, for example, where alveolar cells and intestinal epithelial cells are mixed together (although both are epithelial types, they are quite distinct); and for the brain data, based on the ground truth labels, Harmony actually performs better in keeping subpopulation identities as well as preserving data structure/relationships of cells with similar lineage. These all suggest to me that SCALEX does not perform as well as claimed when the dataset is more complex. While the various scores may be higher for SCALEX, it appears that the biological interpretation of the integrated results is not satisfactory, based on the ground truth labels.

Reviewer #4 (Expertise: scRNASeq integration, comp bio):

In this manuscript, Xiong, Tian et al, develop a new method, SCALEX, for online integration of single-cell datasets based on a variational autoencoder. They achieve this by designing the encoder to preserve only batch invariant signals and adding back batch information in the decoding step via a specific batch normalization layer. Benchmarking against existing state-of-the art methodology shows good performance on a number of integration metrics and highlights the highly scalable nature of online integration methods. They apply their method to a number of different single-cell modalities and existing atlas scale datasets. I believe this work will have utility in the field as scalable integration methods are becoming increasingly necessary as the size of datasets rapidly expands. I have the following specific comments.

Main comments:

1. There is a brief discussion comparing SCALEX to scArches in terms of model augmentation and retraining. I believe this section would benefit from an expanded discussion of how online integration is different from the type of model augmentation that scArches employs.
2. The authors apply SCALEX to multimodal integration (scRNA + scATAC). It would be useful to see a comparison to recent methods tailored towards multi-modal integration like totalVI or Seurat v4.
3. Most examples in the manuscript use datasets of fully differentiated cell types. I would be interested to see how SCALEX performs on data from a developing system. Are there artificial clusters induced or is the developmental continuum preserved?
4. The authors demonstrate that the scope of an existing cell space can be expanded by online projection and demonstrate this by projecting melanoma data onto a previously constructed PBMC space. Does the order of projection here matter and is there a difference between the results if you integrate upfront versus project in afterwards?
5. The latent representation learned by SCALEX is a 10 dimensional bottleneck layer. How was this chosen and is the performance of SCALEX dependent on the choice of this hyperparameter?

Minor comments:

1. The text could be improved with a close read for typos and grammatical errors.
2. On L106, the authors claim that their mini-batch strategy “more tightly follows the overall distribution of input data”. It would be useful to support this claim.
3. There is quite a large reliance on UMAP plots as a way of assessing performance in many of the figures (e.g 2d). I would encourage the authors to try and consolidate or summarize this information in other ways.

Reviewer #5 (Expertise: pulmonary infection, COVID19, single cell):

In this manuscript, Xiong and colleagues provided a deep learning method SCALEX for the integration of heterogeneous single-cell datasets. As dramatic increases in data scale and sample heterogeneity, there is a growing need for integration tools, Therefore, the study is of interest to a large number of readers.

One of the advantages of SCALEX is that it's an online data integration method, which does not require retraining and makes it particularly useful for Atlas-level datasets. SCALEX can also preserve biological variations and avoid over-correction compared to other integration methods. Additionally, SCALEX is designed to preserve batch-invariant biological data components when projecting single-cells. The analysis and comparison of SCALEX with other integration tools showed improved integration accuracy, scalability and computational efficiency.

The manuscript is interesting and very well written. Only a few minor notes:

1. Multiple panels are missing significant analysis.
2. Abbreviation should be spelled out when first used.
3. References were missing for lines 347 and 354.

REVIEWER COMMENTS

Reviewer #1:

In this manuscript, the authors have developed a deep-learning-based online integration method named SCALEX. The authors carried out a rigorous benchmark study with well-established methods in different datasets using different evaluation metrics. Their benchmark study demonstrated that SCALEX has overall better integration performance than other SOTA methods. They also showed that SCALEX has good generalizability and is capable of atlas integration. The unique advantage that SCALEX has over other integration methods is online learning, which can continuously build an integrated reference without retraining when new data arrive. Overall, this is a quality manuscript with substantial results to support its validity. However, some concerns and areas for improvement are discussed below.

Major:

1. The authors present a convincing argument that SCALEX has overall better generalizability than scVI and online iNMF. However, the major focus, “online learning” of SCALEX, is not as well supported and the manuscript needs more appropriate experiments to demonstrate the capabilities of online learning. New pancreas batches and new SKCM batches have a lot in common with the training datasets. Therefore, while projecting new batches into the existing space shows that SCALEX has good generalizability, this is not a challenging enough task to demonstrate that SCALEX is truly “online learning”. Based on my understanding of online learning, the arrival of new data should further fine tune the existing model (learning continues where the entire dataset is not available at any point during training). It is not clear in the manuscript if the authors did this for online iNMF. In the next section, the authors showed that SCALEX is capable of atlas data integration. These atlas datasets can be good examples to demonstrate that SCALEX is indeed an effective online learning method. The authors should perform the atlas data integration in an online manner. For example, the authors could start from one or two tissues across 3 atlas studies, and then fine tune the model with a new tissue. Gradually, the model should reach a similar outcome to the representation obtained by the full atlas data integration. It would be very useful to compare how this online integration result would be different from an atlas integration by SCALEX with all datasets present at one time.

RESPONSE: We now understand the lack of clarity in our originally submitted manuscript. While we do appreciate the reviewer’s comments and suggestions on “online learning”, it must be emphasized that SCALEX is **NOT** an “online learning” method. In fact, SCALEX is an “online” integration method, with *“the ability to incorporate new data without*

recalculating from scratch" (Gao *et al.*, 2021); this means that SCALEX does not "fine tune the existing model" with new incoming data.

While we do expect that "online learning" may improve integration results after fine tuning on new data, the fundamentally enabling design element of SCALEX is that its encoder is **a generalized function** that projects diverse single-cell datasets into the same batch-invariant global cell-embedding space **without retraining**. This design endows SCALEX with a unique advantage: the ability to **align data coming from new single-cell analyses (from the lab and clinic) into the substantial corpus of existing knowledge**, especially that gleaned from the foundational single-cell research efforts, without requiring the current, typical 'recalculating from scratch" approach.

Neither SCALEX nor online iNMF requires retraining (neither is an "online learning" method). We want to emphasize that, as we have shown with multiple tests, SCALEX is more accurate for data integration than online iNMF and than scVI (which is an "online learning" tool that retrains the original model by augmenting new neural network nodes for new data).

2. In the label transferring task, the weighted F1 score reflects the overall accuracy of the model. However, it might not be a good choice to evaluate label transferring for single cell data. Since most single cell data are imbalanced, the macro F1 score can be a better way to evaluate how the model performs for non-dominant cell types. Indeed, the authors show >0.95 F1 scores in Fig. 3c, but the confusion matrices indicate the model is just mediocre for non-major cell types like "mast", "quiescent stellate", and "Schwann". It is not clear whether SCALEX can accurately integrate single cell data that are very imbalanced in their cell type representations. This should either be tested further or discussed as a limitation of SCALEX.

RESPONSE: We thank the reviewer for the guidance here. We have now used the macro F1 score to evaluate the integration performance of different methods on non-dominant cell types (**Revision Figure 1**). Using this index, SCALEX again outperformed online iNMF and scVI, successfully annotating non-major cell types such as mast cells (n=80) and schwann cells (n=62) (left panel); online iNMF and scVI performed quite poorly (middle and right panels).

Revision Figure 1 (the same as Figure 3c in the revised manuscript). Confusion matrix between ground truth cell-types and those annotated by different methods. Left: SCALEX; middle: online iNMF; right: scVI.

3. Again, the good overlap between SC4 consortium and SCALEX COVID-19 PBMC atlas may only suggest good generalizability but not good online learning. Meanwhile, Pearson correlation is not a good metric to support the claim that “the cell-types of two atlases were well-aligned” (Fig. 5i). A confusion matrix after label transferring would better demonstrate the degree of alignment. As above, it seems SCALEX struggles to distinguish non-major DC as shown in Fig. 5i. This again may suggest a limitation that SCALEX is less able to distinguish non-major cell types.

RESPONSE: We appreciate this guidance. First, as we have discussed in the **Response to Point #1**, indeed, the goal of SCALEX is good generalizability but not online learning.

Following the reviewer’s guidance, we have now replaced the Pearson correlation data with new confusion matrix data (**Fig. 5i** and **Revision Figure 2**). Happily, the latter lends strong support to our claims in the manuscript. After projection, the new cell type annotations by label transfer and the original cell types of SC4 Atlas were well-aligned, for instance the original non-major DC_c4-LILRA4 cells were accurately annotated as pDC cells. Of particular note, some DC_c2-CD1C cells and DC_c3-LAMP3 cells were annotated as monocytes. The main reason is that according to the original correlation matrix, mDCs and monocytes have very similar gene expression profiles (which makes sense since they both develop from granulocyte-macrophage progenitors (Buenrostro et al., 2018)). In addition, the locations of mDCs and monocytes in the UMAP embedding are positioned close to each other (**Fig. 5h**).

Revision Figure 2 (the same as Figure 5i in the revised manuscript). Confusion matrix between ground truth cell-types of the SC4 Atlas and those annotated by SCALEX after projecting SC4 Atlas onto the previously constructed SCALEX COVID-19 PBMC Atlas space.

4. Though cross-domain integration is not a focus of this study, the author brought up integrating RNA-seq and ATAC-seq data in Fig. 1f and g. The author should provide a more thorough and quantitative investigation on how SCALEX is competing against the many other existing cross-domain integration methods. Or, if this is not intended to be a major focus, this aspect should perhaps be removed from the study.

RESPONSE: We thank the reviewer for this suggestion. We want to highlight that some of the state-of-the-art scRNA-seq data integration methods examined in our comparative studies are also capable of cross-domain integration by linking features between scRNA-seq and scATAC-seq assays, for example, Seurat v3 (Hao et al., 2021; Zifra et al., 2021), online iNMF (Gao et al., 2021) and Conos (Barkas et al., 2019). We have performed a complete comparison of SCALEX with these integration methods and SCALEX performed exceptionally well (**Supplementary Fig. 7**).

Following the guidance, we have now included two additional cross-domain integration methods, scJoint (Lin et al., 2022) and bindSC (Dou et al., 2020), to our comparisons in the revised manuscript. We noticed that scJoint did not perform well with unpaired scRNA-seq and scATAC-seq datasets: for example, CD4 Naïve, CD8 effector, and Double negative T cells were separated across different batches. On the other hand, bindSC mixed CD4 Naïve and CD8 Naïve cells together. SCALEX substantially outperformed scJoint and bindSC for integration in terms of all examined evaluation scores (**Revision Figure 3**).

Revision Figure 3. Comparisons of integration performance of different methods based on the unpaired *PBMC* dataset. **a**, UMAP embeddings of the unpaired *PBMC* dataset by the indicated methods. Cells are colored by batch or cell-type. **b**, Scatter plot showing a quantitative comparison of different metrics for the unpaired *PBMC* dataset. scJoint and bindSC are highlighted with red circles.

Minor:

1. The human heart data consists of single cells, nuclei and CD45+ enriched cells but the authors only consider addressing batch effects for nuclei data. It would be helpful to demonstrate how SCALEX handles batch effects that arise from how cells are collected.

RESPONSE: We thank the reviewer for this guidance. We have now examined the integration performance of SCALEX on the whole human heart dataset with four batches. We noticed that SCALEX showed excellent performance in removing batch effects between two single-cell batches and two single-nuclei batches; however, the “batch effects” between single-cell and single-nuclei batches were apparently not removed in the UMAP embedding space (**Revision Figure 4a**).

We suspect that these “batch effects” were not true batch effects (*i.e.*, technical noise) but rather represented biological variations; this speculation is based on our detection of a large number of differentially expressed genes when we compared the gene expression profiles between the single-cell and single-nuclei batches. There were 4,015 and 12,855 genes with elevated expression in the single-cell and single-nuclei batches, respectively ($\log\text{-fold change} > 2$ and $\text{adjusted-pvalue} < 0.01$, total gene number = 33,516) (**Revision Figure 4b, c**). We performed gene ontology (GO) enrichment analysis and found that the genes with elevated expression in the single-cell batches showed enrichment for annotations with ribosome-related GO terms (**Revision Figure 4d**). This is reasonable, since single-cell RNA sequencing will (by its nature) detect more cytoplasmic transcripts than single-nuclei RNA sequencing. These results therefore provide additional support for

our conclusion that SCALEX is capable of removing batch effects while retaining true biological differences.

Revision Figure 4. Integration performance of SCALEX on the whole human heart dataset. **a**, UMAP embeddings of the whole *heart* dataset after integration by SCALEX. Cells are colored by cell-type or batch. **b**, Normalized marker gene expression in the UMAP embeddings of the single-cell batches (top) and the single-nuclei batches (bottom). Color bars represent gene expression levels. **c**, Heatmap showing the normalized expression of the top-150 ranking specific genes in each batch. **d**, GO terms enriched in the differentially expressed genes for single-cell (left) or single-nuclei batches (right).

2. Different integration methods may have different properties in terms of preserving data structure locally and globally. The authors did not provide details about how they clustered cells using integrated spaces learned by the different methods. Without these details, there is a concern that clustering parameters may have been chosen that artificially favor SCALEX leading to SCALEX having the highest ARI and NMI in every case. For example, Supplementary Fig 3 suggests that the author over-clustered cell types for other methods but selected a good clustering resolution for SCALEX. More details of how cells were clustered fairly could help clear up this potential appearance or existence of bias.

RESPONSE: We greatly appreciate the reviewer and apologize for not including these details in our originally submitted manuscript. To clarify, we have used the same set of clustering parameters for SCALEX and the other state-of-the-art single-cell integration

tools on the benchmark datasets. We have now included clustering details in the revised manuscript (“Clustering” subsection in **Methods, Supplementary Table 3**, also see **Revision Table 1** below).

For Harmony, MNN, Conos, BBKNN, Scanorama, scVI, LIGER, and online iNMF, we used their latent features with method specific default dimensions for further clustering. For Seurat v3, we initially performed integration and obtained the 2,000-dimensional latent feature vectors following the standard workflow, and then we used PCA for dimensionality reduction because 2,000-dimensional latent feature vectors are too high to directly cluster. Finally, we used 50-dimensional PCA latent feature vectors for clustering. For Conos and BBKNN, since they do not provide latent feature vectors after integration (and we failed to extract the latent feature vectors from their constructed either neighborhood or joint graphs), we used UMAP features for downstream clustering.

To ensure a fair comparison, we used *scanpy.tl.leiden* and *scanpy.tl.louvain* functions for clustering with *resolution=0.5* (**Revision Table 1**). For BBKNN and Conos, since *resolution=0.5* generates too many clusters, we also included clustering results of with *resolution=0.05* for comparison.

In addition, seeking to avoid possible bias from the different dimensions of latent features of different methods, we also used UMAP features (2-dimensional) for clustering and recalculated the NMI-ARI scores for SCALEX and other state-of-the-art single-cell integration tools. SCALEX also performed among the best on benchmark datasets for cell type separation and batch mixing (**Revision Figure 5**).

Revision Table 1. Parameters and cluster numbers of different methods.

	Resolution	Datasets				
		pancreas	heart	liver	NSCLC	PBMC
Raw	0.5	24	24	26	42	18
SCALEX	0.5	9	14	12	17	11
Seurat v3	0.5	12	24	21	27	16
Harmony	0.5	12	19	18	26	16
MNN	0.5	10	16	17	18	11
Conos	0.5	33	74	51	82	31
BBKNN	0.5	35	72	45	72	36
Scanorama	0.5	13	18	17	40	11
scVI	0.5	10	25	19	36	14
LIGER	0.5	17	24	20	20	12
online iNMF	0.5	13	20	20	19	15
Conos	0.05	11	18	19	26	14
BBKNN	0.05	13	23	21	23	14

Revision Figure 5. Scatter plot comparing SCALEX and the other state-of-the-art single-cell data integration tools in terms of the ARI and NMI scores based on Leiden (top) and Louvain (bottom) clustering results.

3. Some typos in the manuscript. For example, line 252: “model retraining” instead of “model retaining”.

RESPONSE: We thank the reviewer for pointing out the typos in our manuscript. We have carefully revised the manuscript and corrected these mistakes.

We would again like to take this chance to sincerely thank the reviewer for the helpful guidance on how to improve our study. Thank you!

Reviewer #2:

The authors provide a method to integrate large scale datasets without requiring retraining and large processing times and computational power. This framework will be useful for curating consensus between different studies and useful for consort where data is not generated in one go but rather generated parallels across labs and tissue types.

Specific questions to address:

1. Throughout the paper, the authors describe superiority of SCALEX in assessing cell types. The metrics used to determine overcorrection help to understand that to a certain extent But the hindrance to biological interpretation is less clear to me. I miss the understanding of whether this is an issue of aesthetics in the UMAP projection or also the cluster assignment itself. For eg, a cell type assessed by Seurat can look split in the 2D embedding, but might have been assigned the same cluster number, thus still preserving

the biological message. Changing clustering parameters to enable that is also a very important task for researchers. It would be useful to elaborate on this when describing the performance metrics.

RESPONSE: We would first like to express our thanks for the supportive comments and the excellent guidance about how to improve our study. Regarding this comment specifically, we agree that the clustering results may sometimes appear differently to the 2D embeddings.

First, for fairness in clustering comparison, we have kept using the same set of clustering parameters for SCALEX and the other state-of-the-art single-cell integration tools on all benchmark datasets. We have now included clustering details in the revised manuscript (“Clustering” subsection in **Methods, Supplementary Table 3**, also see **Revision Table 2** below). Specifically, for Harmony, MNN, Conos, BBKNN, Scanorama, scVI, LIGER, and online iNMF, we used their latent features with method specific default dimensions for clustering. For Seurat v3, we initially performed integration and obtained the 2,000-dimensional latent feature vectors following the standard workflow, and then we used PCA for dimensionality reduction because 2,000-dimensional latent feature vectors are too high to directly cluster. Finally, we used 50-dimensional PCA latent feature vectors for clustering. For Conos and BBKNN, since they do not provide latent feature vectors after integration (and we failed to extract the latent feature vectors from their constructed either neighborhood or joint graphs), we used UMAP features for downstream clustering. We used the *scanpy.tl.leiden* and *scanpy.tl.louvain* functions for clustering with *resolution=0.5*. For BBKNN and Conos, since *resolution=0.5* generates too many clusters, we also included clustering results of with *resolution=0.05*, which were used in our benchmark comparison.

Second, seeking to avoid possible bias from the different dimensions of latent features of different methods, we also used UMAP features (2-dimensional) for clustering and recalculated the NMI-ARI scores for SCALEX and other state-of-the-art single-cell integration tools. SCALEX also performed among the best on benchmark datasets for cell type separation and batch mixing (**Revision Figure 6**).

Revision Table 2. Parameters and cluster numbers of different methods.

		Datasets				
		pancreas	heart	liver	NSCLC	PBMC
Raw	Resolution	24	24	26	42	18
SCALEX	0.5	9	14	12	17	11
Seurat v3	0.5	12	24	21	27	16
Harmony	0.5	12	19	18	26	16
MNN	0.5	10	16	17	18	11

Conos	0.5	33	74	51	82	31
BBKNN	0.5	35	72	45	72	36
Scanorama	0.5	13	18	17	40	11
scVI	0.5	10	25	19	36	14
LIGER	0.5	17	24	20	20	12
online iNMF	0.5	13	20	20	19	15
Conos	0.05	11	18	19	26	14
BBKNN	0.05	13	23	21	23	14

Revision Figure 6. Scatter plot comparing SCALEX and the other state-of-the-art single-cell data integration tools in terms of the ARI and NMI scores based on Leiden (top) and Louvain (bottom) clustering results.

2. In multimodality dataset integration, the authors show integration of scRNAseq and gene expression inferred by scATACseq. Although SCALEX shows a really good embedding of these two different assays, the comparison with existing specific pipelines (apart from supfig7) for such multiomics assays is lacking. How does SCALEX improve that? Plus, advances in mutiomics assays now can enable scATAC and snRNA detection from same cell. Can SCALEX also improve analysis of such datasets?

RESPONSE: We greatly appreciate the reviewer's guidance here. We have now included two additional cross-domain integration methods, scJoint (Lin et al., 2022) and bindSC (Dou et al., 2020), to our comparisons in the revised manuscript. We noticed that scJoint did not perform well with unpaired scRNA-seq and scATAC-seq datasets: for example, CD4 Naïve, CD8 effector, and Double negative T cells were separated across different batches. On the other hand, bindSC mixed CD4 Naïve and CD8 Naïve cells together. SCALEX substantially outperformed scJoint and bindSC for integration in terms of all examined evaluation scores (**Revision Figure 7**).

In addition, we have followed the reviewer's suggestions and tested SCALEX on a *PBMC* multi-omics dataset in which gene expression and chromatin accessibility profiles

are measured in the same cells using the 10X platform (https://support.10xgenomics.com/single-cell-multiome-atac-gex/datasets/1.0.0/pbmc_granulocyte_sorted_10k?). We found that SCALEX performed among the best in terms of all evaluation metrics (**Revision Figure 8a, b**). This supports that SCALEX is capable of integrating scATAC and snRNA data from the same cell.

We also examined the integration performance of totalVI (Gayoso et al., 2021b) and Seurat v4 (Hao et al., 2021) on this *PBMC* multi-omics dataset (**Revision Figure 8c**). These two tools are specifically designed for paired sequencing data integration and cannot be used to integrate data from unpaired cells, and thus cannot be directly compared with the other tools assessed in our study based on matched labels. We therefore did not include them into comparative studies.

Revision Figure 7. Comparisons of integration performance of different methods based on the unpaired *PBMC* dataset. a, UMAP embeddings of the unpaired *PBMC* dataset by the indicated methods. Cells are colored by batch or cell-type. **b**, Scatter plot showing a quantitative comparison of different metrics for the unpaired *PBMC* dataset. scJoint and bindSC are highlighted with red circles.

Revision Figure 8. Comparisons of integration performance of different methods based on the paired *PBMC* dataset from the 10X platform. a, UMAP embeddings of the paired *PBMC* dataset by the indicated methods. Cells are colored by batch or cell-type. **b**, Scatter plot showing a quantitative comparison of different metrics for the paired *PBMC* dataset. **c**, UMAP embeddings of the integration results by Seurat v4 (left) and totalVI (right) based on the paired *PBMC* dataset. Cells are colored by cell-type.

3. The hepatocytes subsets are defined here by single gene differences. How biologically relevant are these subsets? How important is using SCALEX for this tissue type?

RESPONSE: We apologize for not offering details in the originally submitted manuscript. To clarify, we defined the hepatocyte subtypes using a set of differentially expressed genes.

There are normally hundreds of specifically expressed genes in different hepatocyte subsets (**Revision Figure 9a**).

We also would like to note that each hepatocyte subtype showed strong enrichment for annotations with different GO terms (**Revision Figure 9b**), indicating their specific biological significance. We then used single gene with specific biological meanings to label each hepatocyte subset. For example, CXCL1, also known as Gro-alpha, has been recognized as a biomarker associated with hepatocellular carcinoma (Wu et al., 2009), while SCD is related to human fatty liver (Kotronen et al., 2009).

Revision Figure 9. Definition and biological relevance of hepatocyte subtypes. a, Heatmap showing the normalized expression of the top-50 ranking specific genes in each hepatocyte subtype. **b,** GO terms enriched in the specifically expressed genes for different hepatocyte subtypes.

4. Based on fig2b, the overcorrection score looks to show that different techniques are performing differently based on the tissue type. So for example in case of heart data, there was hardly any difference in performance metrics between integration methods. Would you comment on that?

RESPONSE: Overcorrection score is affected by the number of common cell types among different batches in a dataset (*i.e.*, overlapping degree). For the *heart* dataset, all batches include almost the same set of cell types; thus no method suffers from overcorrection. In addition, all cell types in the *heart* dataset are fully differentiated (*i.e.*, are well separated in terms of gene expression profiles), which also makes it an easy dataset with little chance to be overcorrected (*i.e.*, mixing distinct cell types together) (**Supplementary Fig. 2**).

5. Figure 2D is a bit unclear. Extent of overlap as a concept needs to be described a bit more. Also, please attribute the exact subheadings in the methods section when referring to it in the main text. That way we are sure to be looking at the exact relevant subsection. Why I find that this needs more explanation is because the abundance of mixing distinct cell types using other methods is quite a lot I feel. I do not recollect reading such discrepancies in existing literature on single cell atlas in pancreas.

RESPONSE: We are sorry for the unclear description of the extent of overlap in our originally submitted manuscript. We agree that it is very often that integration methods mix distinct cell types (frequently because of this partial-overlap issue). We appreciate the reviewer's clarifying suggestions. In fact, the extent of overlap between two batches is the number of common cell types between two batches. For example, *overlap_2* means there are two common cell types (*e.g.*, alpha and beta) between the *pancreas_celseq* and *pancreas_smartseq2* batches. We have taken care to describe the concept and the generation of partially overlapping datasets in the methods subsection "**Generation of partially overlapping datasets**".

In addition, we have carefully checked our manuscript to ensure that the exact subheadings in the methods section are referred in the main text.

6. On similar lines, intermediate polyhormonal cells in endocrine pancreas might be of interest. Is there a possibility that SCALEX, as it relies on existing knowledge of cell labels or markers, can miss out on them?

RESPONSE: We thank the reviewer for focusing our attention on these cells. We reasoned that SCALEX did not identify intermediate polyhormonal cells in endocrine pancreas is

because there were no such cells in the benchmark *pancreas* dataset we used in our study. To demonstrate, we have now collected a new scRNA-seq dataset of vitro β -cell differentiation of human (Veres et al., 2019). In that study, the authors identified a new cell type called SC-alpha cells which expressed not only the markers of islet alpha cells but also insulin. The authors thought they were intermediate poly-hormonal cells.

We projected these SC-alpha cells into the pancreas cell space using the same SCALEX encoder trained on the *pancreas* dataset. We noticed that SCALEX did not project these cells onto any existing cell populations in the pancreas space; rather, SCALEX projected these cells onto new locations close to alpha and beta cells (**Revision Figure 10a**). We re-examined the gene expression of *INS* and *GCG*, which are marker genes of beta cells and alpha cells, respectively. Indeed, the projected SC_alpha cells highly expressed both *INS* (Insulin) and *GCG* (Glucagon) (**Revision Figure 10b**), which is consistent with the conclusions from the aforementioned study (Veres et al., 2019). In other words, if there had been such intermediate polyhormonal cells as a group in the existing cell space, we would have been able to discover them.

Revision Figure 10. Projecting intermediate poly-hormonal cells into pancreas space. **a**, UMAP embeddings of the common cell space that includes the original *pancreas* dataset (left) and the projected intermediate poly-hormonal cells (right). Cells are colored by cell-type. **b**, Normalized *INS* (left) and *GCG* (right) expression on the UMAP embeddings. Color bar represents the expression level.

7. For figure 3E, this location of tumor cells, would it change with multiple heterogenous tumor samples? Is this distance between clusters that informative? Usually these distances can be very tricky to interpret. Could you comment on that?

RESPONSE: Thank you for inviting us to consider (and speculate on) this topic. First, for tumor cells from multiple heterogenous samples, our experience is that if they have quite different gene expression profiles (because of distinctively disrupted gene regulation programs), they would be projected onto different locations.

Second, we agree that the distances between clusters in UMAP embeddings are very tricky to interpret, as they are uninformative in theory.

8. For the novel cell type determined in pancreas atlas, is there any particular technique or sample from which these cells come from? Or are they present in multiple libraries? SLC16A7 or MCT2 expression has been investigated before in the context of islet function, but remained undetected (Zhao et al. (2001). Expression and Distribution of Lactate/Monocarboxylate Transporter Isoforms in Pancreatic Islets and the Exocrine Pancreas. Diabetes) What can be the reason that SCALEX identifies these cells? Can they be infiltrating cells? Again, does the guided labelling in the reference atlas impact the projection of such non-overlapping datasets?

RESPONSE: The newly found SLC16A7+ epithelial cells (n=50) are mainly from one dataset, the *pancreas_celseq* dataset (n=41). We thought that these cells are too rare to be detected by bulk sequencing or microscopic immunohistochemical imaging. In contrast, single-cell sequencing approaches are suited for identifying such rare cell types by deconvolving heterogeneous cell populations. SCALEX has the great ability to retain subtle biological differences among cells in single-cell sequencing data during integration, and we found that these uncharacterized cells were well-separated from other cell groups after SCALEX integration.

Indeed, there is to date no evidence that any pancreatic cell types express SLC16A7 or MCT2. Moreover, as the vast majority of these cells are from one study, suggesting that these cells may be infiltrating cells rather than a rare subtype of “real” rare pancreatic cells.

Finally, we note that the projection of SCALEX is independent of the labels of the reference data for all cell types, including the non-overlapping cell types, which will be projected onto new locations in the reference cell space.

9. COVID atlas looks very interesting! Have the authors used the PBMC atlas mentioned in earlier sections of the paper?

RESPONSE: We did not integrate the *PBMC* dataset (involved in the benchmark datasets) in earlier sections with COVID-19 PBMC Atlas in our originally submitted manuscript. We have here now examined integrating the COVID-19 Atlas and the *PBMC* dataset. The

same cell types between these two datasets were well mixed while different cell types were well separated (**Revision Figure 11**), which again highlights the excellent integration performance of SCALEX.

Revision Figure 11. Integration performance of SCALEX on the COVID-19 PBMC Atlas and the *PBMC* dataset. UMAP embeddings of the COVID-19 PBMC Atlas (left) and the *PBMC* dataset (right) after SCLAEX integration. Cells are colored by cell-type.

10. In figure 5g, where the frequencies are described, is it % of total cells in that sample? Without normalising for total number of cells, such fluctuating numbers can be tricky esp in diseased states.

RESPONSE: Yes, the frequency described in Fig. 5g is the percentage of each cell type in each sample; it has been normalized, thus enabling comparisons amongst different samples.

11. So basically for atlases like covid and cancer, where knowledge of cell types and states is very recent, how will SCALEX perform in comparison to just looking into smaller subsets with more directed research questions? Could you comment on its strengths and drawbacks if any?

RESPONSE: As we understand, the reviewer is here interested in having us discuss the strengths and drawbacks for large-scale atlas studies and smaller-scale focused studies using SCALEX as an analytical tool. Regarding atlas studies, they typically aim to provide a global overview of the cell types and states. In addition to the obvious drawback of high-cost, another challenge for atlas studies is the difficulty in integrative data analysis (Luecken et al., 2022). Thus, we are particularly happy present our evidence showing how SCALEX can be used to informatively dissect biological consensus and variations among different conditions, donors, stages of disease, etc., from technical factors that are often highly heterogenous in large-scale, collaborative atlas studies. For example, we used SCALEX to integrate multiple COVID-19 studies and built a COVID-19 PBMC Atlas with

diverse disease states, SCALEX identified multiple immune cell subpopulations such as monocytes, neutrophil and plasma were differentially associated with patient status. Such discovery is difficult or even impossible for small-scale studies because of limited sampling of various disease states.

Focused studies, in contrast, are appropriate when addressing a focused research question, and typically take care to controlled biologically conditions (to properly isolate the specific variable under study). While focused studies are obviously cheaper, they are (by design) limited to certain conditions (e.g., genetic background). In this regard, it is fortunate to find that SCLAEX is effective in supporting unbiased integration of many focused studies: that is, a scientist studying a given research field can use SCALEX to do informative analyses that harness the data from any/all focused studies about that topic (whenever they were produced). SCALEX's unbiased integration allows researchers to expand the biological scope of their investigations, increasing coverage to include multiple materials/conditions (and the associated variance) Nevertheless, we note that it remains early days for integration efforts of multiple focus studies, and this is more challenging than integrating atlas datasets from a single (even large) study; and there is a danger for biased observations if the set of focused studies examined contain bias towards particular conditions.

In conclusion, as my own expertise is closer to extracting biologically relevant points from these single cell investigations, I feel that the authors should emphasis the role of SCALEX in facilitating that. In terms of processing, I am convinced that it is very advantageous and shows promise. Discussing the advantages in biological consensus and translatability would be nice.

RESPONSE: We thank the reviewer for the guidance and we have carefully revised the manuscript to emphasize the role of SCALEX in facilitating extracting biologically relevant points from single cell investigations. For example, in the task of integrating partially overlapping dataset of *liver*, SCALEX is the only tool that correctly maintained the five hepatocyte subtypes apart with specific biological significance (**Fig. 2a** and **Revision Figure 9**). We have also improved our discussion regarding the advantages of SCALEX in terms of biological consensus and translatability. For example, by integrating multiple COVID-19 studies to build a COVID-19 PBMC Atlas with diverse disease states, SCALEX identified multiple immune cell-types such as monocytes, neutrophil, and plasma that were closely related to COVID-19 disease progression across multiple small-scale datasets.

We greatly appreciate the reviewer's supportive comments and would again like to take this chance to sincerely offer our gratitude for the helpful guidance about how to improve our study. Thank you!

Reviewer #3:

The authors have reported a single-cell data integration framework called SCALEX, which fundamentally uses a VAE, with a domain-specific batch normalization approach to align different single-cell RNA-seq datasets. They claim that their method does not require re-training on new datasets, and that as such their method can be deployed online with faster and more accurate performance than other existing methods. Although I appreciate the utility of being able to analyze large-scale single cell data online, I have doubts as to the extent of novelty of the method given the many other existing methods already in the space. In addition, I am not convinced that their method is more accurate than others out there sufficiently so to overlook the lack of novelty in the method itself. My specific major comments are below:

Major comments:

1. The SCALEX method is built upon the VAE framework. First, the authors themselves have previously reported this autoencoder for single-cell ATAC-seq data analysis: "We previously applied VAE and designed SCALE (Single-Cell ATAC-seq Analysis via Latent feature Extraction) to model and analyze single-cell ATAC-seq data". The majority of the novelty in using this type of framework for extracting latent features from single-cell data has already been reported in their previous publication. The only difference between the previous publication and the one being considered at present is the proposed domain-specific batch normalization (DSBN) in a batch-free encoder and batch-specific decoder, respectively. This is, in my opinion, not very novel as the specific normalization method is not new and only represents a minor change to SCALE.

RESPONSE: We think that as a method, the novelty of SCALEX lies in the three design elements:

- 1) an asymmetric autoencoder that inputs batch information only to the decoder (*i.e.*, never to the encoder);
- 2) a DSBN layer in the decoder to release the encoder from the burden of capturing the batch-specific variations;

3) a mini-batching strategy that samples data from all batches instead from a single batch and thus more tightly follows the same overall distribution of the input data; this strategy includes a Batch Normalization layer in the encoder that adjusts the deviation of each mini-batch and align them to the overall input distribution.

It bears special emphasis that its design elements, when functioning together, render SCALEX's encoder as **a generalized function** that enables projection of diverse single-cell datasets into the same batch-invariant global cell-embedding space. As we have shown with our extensive "feature ablation" studies that the removal of any design element leads to a substantial performance drop in the accuracy of the generalized projection function. **We would like to note that this concept of generalized projection function (or a batch-invariant global cell-embedding space) is highly innovative:** other than SCALEX, only online iNMF (recently published on Nature Biotechnology) is able to project new incoming cells into an existing global cell-embedding space, and our data demonstrate that SCALEX substantially outperforms online iNMF in terms of projection accuracy.

The benefits of such a generalized projection function are tremendous. We here summarize them as three novel, major advantages for an integration tool:

1) SCALEX is **a truly online integration method**. Only SCALEX and online iNMF are capable of online data projection or integration. Note that here "online" represents "*the ability to incorporate new data without recalculating from scratch*" (Gao *et al.*, 2021). We want to highlight that this online integration ability meets a rapidly growing need in the life sciences and in biomedicine: it enables the alignment of data coming from new single-cell analyses (from the lab and clinic) into the substantial corpus of existing knowledge from foundational single-cell research.

2) SCALEX is unique in its capacity to **accurately integrate partially-overlapping datasets**, where all other integration tools performed quite poorly. As more and more single-cell analyses are completed for very diverse conditions, there is an increasing need for the ability to integrate these datasets which are very often partially overlapped (*i.e.*, contain non-overlapping cell populations in each batch). Our analyses showed that the other state-of-the-art integration tools often suffered from an over-correction problem when integrating these partially-overlapping data—specifically by merging different cell-types together, when integrating these partially-overlapping data (see **Figure 2** for examples).

3) The autoencoder architecture of SCAELX makes it very computationally efficient and thus **capable of integrating "Atlas-level" datasets**, which are characterized by very large data-size (with recent examples exceeding ~1 million cells, *e.g.*, the Human Fetal Atlas, the COVID-19 Atlas), multiple data-batches generated across distinct conditions, and very heterogenous and complex samples. As from the preprint suggested by this reviewer, these "Atlas-level" datasets pose challenges for data integration with difficulties

of “*the complex, nonlinear, nested batch effects*”. It is clearly an ever-more crucial feature as these “Atlas-level” studies continue to become more and more prevalent.

Beyond the unique feature set for data integration, we also would like to highlight **the excellent data integration performance of SCALEX**. Indeed, the overall data-integration performance of SCALEX is head-and-shoulders above the other state-of-the-art single-cell data integration tools, based on extensive benchmarking and multiple evaluation metrics (**Supplementary Fig. 4**). It bears emphasis that SCALEX suffered the least from the over-correction problem, as assessed by an “over-correction” score.

Furthermore, I have concerns as to the applicability of this method: as BN only shifts and scales data within a mini-batch of 64 cells (arguably in the context of atlas sized datasets it is not a representative sample size even), it presumably has limited ability to remove batch effects; it is unclear why the cell-embedding space of SCALEX should be batch-invariant, based on the method presented, as the theoretical model does not support such an assumption. Can the authors explain/give more intuition on that?

RESPONSE: We respectfully note that studies have shown that usually the size of a mini-batch does not need to be large to achieve excellent model performance (Bengio, 2012; Dominic Masters, 2018). In theory, there is no evidence to support that one mini-batch should cover all of the samples or of the categories. In fact, the selection of a mini-batch size should seek to achieve a balance between the robustness and efficiency of the gradient descent. As for the SCALEX, it is not a problem that each mini-batch is only a partial observation from the total distribution of cell populations; these partial observations will converge in the end to the global distribution in the shared space based on the substantial optimizations conducted in the SCALEX workflow.

In practice, we have tested the performance of SCALEX with the choice of mini-batch size ranging from 16, 64, to 256, and found that the performance of SCALEX was quite robust to different batch sizes. We found that the mini-batch size of 64 is the best choice in terms of integration performance and computation efficiency for large-scale datasets including millions of single cells.

The batch-invariant characteristics of the SCALEX cell-embedding space is the direct consequence of the generalizability of the SCALEX encoder. To achieve this, SCALEX includes three specific design elements described above. We note that in addition to the DSN and the mini-batch strategy, supplying batch information only to the decoder focuses the encoder exclusively on learning the batch-invariant biological components and is crucial for the encoder generalizability. We have conducted extensive “feature ablation”

studies to show that three design elements together make the SCALEX encoder a generalized function for single-cell data integration and projection.

2. Most single-cell integration tools now demonstrate not only ability to integrate single-cell RNA-seq data, but also single-nuclei RNA-seq data. Due to the challenges associated with processing different sample and tissue types, single-cell atlas efforts now routinely use snRNA-seq to profile certain tissues/organs to avoid bias, while using scRNA-seq to profile other tissue types in the same atlas. Adipose and Neuronal tissues are prominent examples that call for snRNA-seq. Given that the authors claim to be superior for atlas-level datasets, and given that numerous other tools have been published in the same space, it would be very important for them to show applicability of their method to integration of not just scRNA/scRNA or snRNA/snRNA, but also scRNA/snRNA. The authors have integrated 'Harvard-Nuclei' and 'Sanger-Nuclei' data from the adult human heart atlas (<https://www.heartcellatlas.org/>). Besides these datasets, the atlas project also provided other two data sources: 'Sanger-Cells', 'Sanger-CD45'. If the authors can show integration of all of these from the same atlas, or at least align 'Sanger-Cells' with 'Sanger-Nuclei', it will demonstrate a more meaningful application of their method (i.e. building alignment between single-cell, single-nucleus data from a single atlas dataset).

RESPONSE: We thank the reviewer for this guidance. We have now examined the integration performance of SCALEX on the whole adult human heart atlas with four batches. We noticed that SCALEX showed excellent performance in removing batch effects between two single-cell batches and those between two single-nuclei batches; however, the "batch effects" between single-cell and single-nuclei batches were apparently not removed in the UMAP embedding space (**Revision Figure 12a**).

However, we suspect that these "batch effects" were not true batch effects (i.e., technical noise) but rather represented biological variations; this speculation is based on our detection of a large number of differentially expressed genes when we compared the gene expression profiles between the single-cell and single-nuclei batches. There were 4,015 and 12,855 genes with elevated expression in the single-cell and single-nuclei batches, respectively ($\log\text{-fold change} > 2$ and $\text{adjusted-pvalue} < 0.01$, total gene number = 33,516) (**Revision Figure 12b, c**). We performed gene ontology (GO) enrichment analysis and found that the genes with elevated expression in the single-cell batches showed enrichment for annotations with ribosome-related GO terms (**Revision Figure 12d**). This is reasonable, since single-cell RNA sequencing will (by its nature) detect more cytoplasmic transcripts than single-nuclei RNA sequencing. These results therefore

provide additional support for our conclusion that SCALEX is capable of removing batch effects while retaining true biological differences.

Revision Figure 12. Integration performance of SCALEX on the whole human *heart* dataset. **a**, UMAP embeddings of the whole *heart* dataset after integration by SCALEX. Cells are colored by cell-type or batch. **b**, Normalized marker gene expression in the UMAP embeddings of the single-cell batches (top) and the single-nuclei batches (bottom). Color bars represent gene expression levels. **c**, Heatmap showing the normalized expression of the top-150 ranking specific genes in each batch. **d**, GO terms enriched in the differentially expressed genes for single-cell (left) or single-nuclei batches (right).

3. The authors have carefully examined the design of SCALEX via an ablation study (Figs. S19, S20). Both BN and DSBN layers show clear impact on integration results. One major concern here is that the results from simple autoencoder are already not bad because the data preprocessing steps have the ability to adjust for batch effects. It appears possible that the data preprocessing already plays a significant role to remove the batch effects, and it is unclear how much of the correction is actually owing to the DSBN. It will be more attractive and convincing to users if the ablation study could be conducted based on more challenging scenarios.

RESPONSE: We greatly appreciate the reviewer for the suggestion about improving our ablation study. First, we want to clarify that the autoencoder involved in our comparison (**Supplementary Fig. 19** and **Supplementary Fig. 20**) was not a simple one but already

combines with Batch Normalization and DSBN. Second, to further investigate the contributions of each SCALEX design element, we have examined the performance of three additional (more challenging) test-variants: Autoencoder without BN, Autoencoder without DSBN, and Autoencoder without BN or DSBN.

The integration and projection performances of these three test-variants were significantly worse than full SCALEX, and were among the worst of all SCALEX test-variants (**Revision Figure 13, 14**), highlighting the importance of the design elements for excellent integration and projection performance of the full SCALEX.

Revision Figure 13. Ablation studies of different SCALEX test-variants for single-cell data integration. a, UMAP embeddings after integration by SCALEX and other SCALEX test-variants. b, Scatter plot showing a quantitative comparison of the performance of the full and different test-variants of SCALEX using the different metrics across the indicated benchmark datasets.

Revision Figure 14. Ablations studies of different SCALEX test-variants for single-cell data projection. Left: UMAP embeddings of three projected pancreas data batches projected onto the pancreas space using the indicated SCALEX test architectures. Right: UMAP embeddings of the two projected melanoma data batches projected onto the PBMC space using different SCALEX architectures. Cells are colored by cell-type. Light gray shadows represent the original embedding space.

Minor comments:

1. Somewhat relevant to point 3 above, most of the datasets selected by the authors (especially in the main figures) appear to be relatively less complex datasets, consisting mostly of tissues that harbor fewer and more distinct cell types, e.g. PBMCs, liver. However, the datasets that are more complex and involve multiple tissue types clearly do not show superior performance. For example, in Fig 4 the entire mouse atlas was integrated, including different technology types. While at first glance the results look ok, with closer examining we find cell types that do not belong together to be mixed, e.g. the neurons and cardiac muscle cells; some of the T cells are mixed with B cells; some cells unknown what type but all mixed together around the skeletal muscles and cardiac cells. In addition, some cells that belong together or nearby are no longer associated, including oligodendrocytes

and their precursors, which are of the same lineage. The situation is more severe in S5 and S6, for example, where alveolar cells and intestinal epithelial cells are mixed together (although both are epithelial types, they are quite distinct); and for the brain data, based on the ground truth labels, Harmony actually performs better in keeping subpopulation identities as well as preserving data structure/relationships of cells with similar lineage. These all suggest to me that SCALEX does not perform as well as claimed when the dataset is more complex. While the various scores may be higher for SCALEX, it appears that the biological interpretation of the integrated results is not satisfactory, based on the ground truth labels.

RESPONSE: Thank you for focusing our attention on this issue; we have now carefully re-checked the integration results of SCALEX on complex datasets. We found that the issues reported by the reviewer could be classified into three categories:

- 1) Visualization problem caused by low-resolution UMAP plottings . For example, the neuron cells and cardiac muscle cells are well-separated in a UMAP plot of higher resolution (**Revision Figure 15a**). Similarly, alveolar cells and intestinal epithelial cells are also well-separated in a high-resolution UMAP plot (**Revision Figure 15e**).
- 2) Annotation and label problems. This includes the case of mixing some T cells with B cells. In fact, we found that cells in cluster 7 which were annotated as T cells in the original paper highly express B cell clinical markers (**Revision Figure 15d**). We thus suspect that these cells are actually B cells, but may have been annotated in error in the original study.
Regarding the case of the “unknown” cells being mixed together around the skeletal muscles and cardiac cells: we must clearly note that in the figures we only labeled the major cell types to enable a clear visual presentation. This choice to limit the number of labeled cell types does (obviously) reduce the resolution available for readers, missing labeling those cells with a small population. If switching to a higher resolution UMAP plot, we can see that these “unknown” cells include basal of epidermis, mesenchymal, some ovary subtypes, etc., and they are not mixed together (**Revision Figure 15c**).
- 3) Potential biological reasons. For the separation of oligodendrocytes and their precursors in the UMAP plot, we note that even though these two cell types are from the same lineage, they respectively contain 131 and 318 differentially expressed genes when compared to each other ($\log\text{-fold-change} > 1$ and $\text{adjusted-pvalue} < 0.05$. #total genes=2,000) (**Revision Figure 15f**), indicating quite different gene expression profiles for these two cell types.

We thus conclude that actually SCALEX performs well on complex datasets. We also

note that for this complex mouse brain dataset, Harmony even didn't align the same cell types together, for example, L2/3 IT, L4, Oligo cells, etc. (**Supplementary Fig. 6**). It is unfair to conclude that Harmony performs better than SCALEX.

Revision Figure 15. UMAP embeddings of neuron cells and cardiac muscle cells (**a**), their nearby cells (**b**), and Ovary cells with their original annotations (**c**) in the Mouse Atlas dataset. **d**, UMAP embeddings of B cells and T cells in the Mouse Atlas dataset. Cells are colored by cell labels (left) and Leiden clustering results (middle). Right: dotplot of canonical marker genes for each cluster. Dot color represents average expression level, and dot size represents the proportion of cells in the group expressing the marker. **e**, UMAP embeddings of alveolar cells and intestinal epithelial cells in the Human Fetal Atlas dataset. **f**, Heatmap showing the normalized expression of the differentially expressed genes in different cell types.

We would again like to take this chance to sincerely thank the reviewer for the helpful guidance on how to improve our study.

Reviewer #4:

In this manuscript, Xiong, Tian et al, develop a new method, SCALEX, for online integration of single-cell datasets based on a variational autoencoder. They achieve this by designing the encoder to preserve only batch invariant signals and adding back batch information in the decoding step via a specific batch normalization layer. Benchmarking against existing state-of-the art methodology shows good performance on a number of integration metrics and highlights the highly scalable nature of online integration methods. They apply their method to a number of different single-cell modalities and existing atlas scale datasets. I believe this work will have utility in the field as scalable integration methods are becoming increasingly necessary as the size of datasets rapidly expands. I have the following specific comments.

Main comments:

1. There is a brief discussion comparing SCALEX to scArches in terms of model augmentation and retraining. I believe this section would benefit from an expanded discussion of how online integration is different from the type of model augmentation that scArches employs.

RESPONSE: We greatly appreciate this helpful guidance. We have now provided a more informative discussion comparing SCALEX with scArches (see page 3 line 70~82), which is copied below:

“Another recently developed package, scvi-tools(Gayoso et al., 2021a), combining scVI(Lopez et al., 2018) with scArches(Lotfollahi et al., 2020), applies a conditional variational autoencoder (VAE)(Kingma and Welling, 2013) framework to model the inherent distribution of the input single-cell data for data integration. However, the conditional VAE design of scVI requires model augmentation and retraining when integrating new data, meaning that scVI is not an online method. We want to highlight that this online integration ability meets a rapidly growing need in the life sciences and in biomedicine: it enables the alignment of data coming from new single-cell analyses (from the lab and clinic) into the substantial corpus of existing knowledge, especially that from previous foundational single-cell research. Put another way, the online integration capacity obviates the need to augment and/or retrain models when analyzing additional datasets, which both preserves hard-won scientific insights and saves a huge amount of computational resource.”

2. The authors apply SCALEX to multimodal integration (scRNA + scATAC). It would be useful to see a comparison to recent methods tailored towards multi-modal integration like totalVI or Seurat v4.

RESPONSE: We thank the reviewer for this suggestion. We have now included totalVI and Seurat v4 and performed extensive comparative analyses on more multi-modal datasets. Note that since totalVI and Seurat v4 integrate multi-omics data measured **in the same cell**, we specifically collected a new PBMC dataset in which the gene expression and chromatin accessibility profiles are measured in the same cell using the 10X platform (https://support.10xgenomics.com/single-cell-multiome-atacseq/datasets/1.0.0/pbmc_granulocyte_sorted_10k?).

We found that SCALEX performs among the best in terms of all evaluation metrics, separating different cell types well while mixing different batches well (**Revision Figure 16 a, b**). These two tools are specifically designed for paired sequencing data integration and cannot be used to integrate data from unpaired cells, and thus cannot be directly compared with the other tools assessed in our study based on matched labels. We therefore did not include them into comparative studies.

Revision Figure 16. Comparisons of integration performance of different methods based on the paired *PBMC* dataset from the 10X platform. **a**, UMAP embeddings of the paired *PBMC* dataset by the indicated methods. Cells are colored by batch or cell-type. **b**, Scatter plot showing a quantitative comparison of different metrics for the paired *PBMC* dataset. **c**, UMAP embeddings of the integration results by Seurat v4 (left) and totalVI (right) based on the paired *PBMC* dataset. Cells are colored by cell-type.

3. Most examples in the manuscript use datasets of fully differentiated cell types. I would be interested to see how SCALEX performs on data from a developing system. Are there artificial clusters induced or is the developmental continuum preserved?

RESPONSE: We greatly appreciate this prompt to examine the integration performance of SCALEX for a developing system. Note that datasets comprising different batches from developing systems are quite rare; we only found one such human immune dataset (three batches of one study) (Oetjen et al., 2018). We manually selected a set of cells belonging to the erythrocyte lineage (HSPCs, Megakaryocyte progenitors, Erythroid progenitors, and Erythrocytes) for testing.

Seeking to define the erythrocyte development trajectory from HSPCs—via megakaryocyte progenitors and erythroid progenitors—to erythrocytes and to assess the integration performance of SCALEX on a developmental continuum, we used the *sc.tl.dpt* function in the *Scanpy* package to compute the diffusion pseudotime scores of 1) unintegrated data and 2) SCALEX integrated data. Note that we assumed that the trajectories defined in the original unintegrated data are accurate. We noted a branching structure in the trajectory constructed from SCALEX integrated data (**Revision Figure 17**), and the Pearson coefficient between the pseudotime values before and after integration is 0.79. These results indicate that SCALEX integration can preserve the trajectory structure of for developing systems.

We would like to note that it appears that the batch effects in this dataset seem small; yet this was the only dataset we could find comprising different batches from a developing system.

Revision Figure 17. Integration performance of SCALEX on an erythrocyte lineage dataset. UMAP embeddings of the erythrocyte lineage dataset before (top) and after integration by SCALEX (bottom), cells are colored by batch, cell-type, diffusion pseudotime of unintegrated data, and diffusion pseudotime of SCALEX integrated data.

4. The authors demonstrate that the scope of an existing cell space can be expanded by online projection and demonstrate this by projecting melanoma data onto a previously constructed PBMC space. Does the order of projection here matter and is there a difference between the results if you integrate upfront versus project in afterwards?

RESPONSE: We thank the reviewer for this guidance, and have now tested the effects of the order of projection. In the new test, we projected the *PBMC* dataset onto the *melanoma* cell embedding space. We noticed that again SCALEX accurately projected cells in the *PBMC* dataset onto the same locations in the cell space of the *melanoma* dataset (**Revision Figure 18**), suggesting that the projection order does not affect the online projection performance of SCALEX.

Revision Figure 18. Projecting new data into an existing cell-embedding space. **a**, UMAP embeddings of the cell space of the *PBMC* dataset, with the two *melanoma* data batches projected using SCALEX. Cells are colored by cell-type, with light gray shadows representing the original *PBMC* dataset. **b**, UMAP embeddings of the cell space of the *melanoma* dataset, with the two *PBMC* data batches projected using SCALEX. Cells are colored by cell-type, with light gray shadows representing the original *melanoma* + dataset.

5. The latent representation learned by SCALEX is a 10 dimensional bottleneck layer. How was this chosen and is the performance of SCALEX dependent on the choice of this hyperparameter?

RESPONSE: Our choice here was guided by extensive hyper-parameter testing (grid search) from the development of our previous model SCALE (Xiong et al., 2019). Across multiple choices of dimensionalities, we found that a latent space dimensionality of 10 generally performed best. As SCALE and SCALEX are both based on a similar autoencoder framework and are both designed for single-cell sequencing data, we adopted the same dimensionality setting of 10.

Minor comments:

1. The text could be improved with a close read for typos and grammatical errors.

RESPONSE: We are thankful for the detailed guidance here. We have carefully revised the manuscript and corrected typos and grammatical errors. For example, “retaining” to “retraining” on line 252, “SCALE” to “SCALEX” on line 1233, etc.

2. On L106, the authors claim that their mini-batch strategy “more tightly follows the overall distribution of input data”. It would be useful to support this claim.

RESPONSE: In the manuscript, we say that “mini-batch sampling from mini-batch strategy that samples data from all batches (instead of single batches)”. We have now modified the text to minimize understanding: “*the SCALEX encoder employs a mini-batch strategy that samples data from all batches (instead of a single batch), which more tightly follows the overall distribution of the input data.*”

To demonstrate that our mini-batch strategy does indeed more tightly follow the overall distribution of input data than the alternative strategy that samples from a single batch, we used the *pancreas* dataset (8 batches) and randomly selected two genes (ASS1 and HSPA1A) to compare their expression variances in the two sampling strategies.

For a mini-batch strategy that samples from all batches, we randomly selected 64 cells from all batches as a mini-batch, and then calculated the variance of each gene among the sampled cells with respect to the variance of that gene among all cells. We repeated this process 8 times. For the strategy that samples from a single-batch, we randomly selected 64 cells of each batch and similarly calculated the variance of each gene among mini-batch cells with respect to the variance of that gene among all cells.

We found that the relative variances of gene expression using our mini-batch sampling strategy were indeed closer to zero than that using the alternative strategy that samples from a single batch (**Revision Figure 19**), supporting our claim in the manuscript.

Revision Figure 19. Gene expression variances comparison between strategies that sample cells from all batches (“all”) or sample cells from a single-batch (“single”). Left: ASS1; right: HSPA1A.

3. There is quite a large reliance on UMAP plots as a way of assessing performance in many of the figures (e.g 2d). I would encourage the authors to try and consolidate or summarize this information in other ways.

RESPONSE: We thank the reviewer for this suggestion. We provided UMAP plots whenever possible, hoping that they could be helpful to provide a full image of the integration performance. We understand the limitation of UMAP plots as a visualization approach, and indeed we used a multitude scoring metrics to summarize the UMAP visualizations and to quantitatively compare integration performance of different methods; this include the use of the Silhouette score, the batch entropy mixing score, the cLISI and iLISI scores, etc., to quantify cell-type separation and batch mixing, ARI and NMI to assess cell-type clustering. We also defined a novel over-correction score to measure the over-correction degree. Using these metrics together with UMAP plots, we strive to provide a comprehensive comparison of integration performance of different methods.

We would again like to take this chance to sincerely thank the reviewer for the helpful guidance on how to improve our study.

Reviewer #5:

In this manuscript, Xiong and colleagues provided a deep learning method SCALEX for the integration of heterogeneous single-cell datasets. As dramatic increases in data scale and sample heterogeneity, there is a growing need for integration tools, Therefore, the study is of interest to a large number of readers.

One of the advantages of SCALEX is that it's an online data integration method, which does not require retraining and makes it particularly useful for Atlas-level datasets. SCALEX can also preserve biological variations and avoid over-correction compared to other integration methods. Additionally, SCALEX is designed to preserve batch-invariant biological data components when projecting single-cells. The analysis and comparison of SCALEX with other integration tools showed improved integration accuracy, scalability and computational efficiency.

The manuscript is interesting and very well written. Only a few minor notes:

1. Multiple panels are missing significant analysis.

RESPONSE: We have now carefully checked and provided all necessary significance analyses in the revised manuscript. For example, we added a significance analysis of cytokine and inflammatory scores of monocytes among different conditions to support our claim on line 392.

2. Abbreviation should be spelled out when first used.

RESPONSE: We thank the reviewer for the detailed guidance here, and we have now carefully checked and spelled out all abbreviations in the revised manuscript. For example, single-cell RNA sequencing (scRNA-seq), single-cell assay for transposase-accessible chromatin use sequencing (scATAC-seq), Single nucleus assay for transposase-accessible chromatin using sequencing (snATAC-seq), cellular indexing of transcriptomes and epitopes by sequencing (CITE-seq), gigabytes (GB), million (M), kilo (K), central processing unit (CPU), graphics processing unit (GPU), gene ontology (GO).

3. References were missing for lines 347 and 354.

RESPONSE: Thanks for bringing these omissions to our notice; we have now added the references in our revised manuscript.

Line 347 (line 364 in the revised manuscript): (Schulte-Schrepping et al., 2020)

Line 354 (line 374 in the revised manuscript): (Zhang et al., 2020)

We would again like to take this chance to sincerely thank the reviewer for the helpful guidance on how to improve our study.

Reference

Barkas, N., Petukhov, V., Nikolaeva, D., Lozinsky, Y., Demharter, S., Khodosevich, K., and Kharchenko, P.V. (2019). Joint analysis of heterogeneous single-cell RNA-seq dataset collections. *Nat Methods* 16, 695-698.

Bengio, Y. (2012). Practical recommendations for gradient-based training of deep architectures. arxiv.

Buenrostro, J.D., Corces, M.R., Lareau, C.A., Wu, B., Schep, A.N., Aryee, M.J., Majeti, R., Chang, H.Y., and Greenleaf, W.J. (2018). Integrated Single-Cell Analysis Maps the Continuous Regulatory Landscape of Human Hematopoietic Differentiation. *Cell* 173, 1535-1548 e1516.

Dominic Masters, C.L. (2018). Revisiting small batch training for deep neural networks. arxiv.

Dou, J., Liang, S., Mohanty, V., Cheng, X., Kim, S., Choi, J., Li, Y., Rezvani, K., Chen, R., and Chen, K. (2020). Unbiased integration of single cell multi omics data. bioRxiv.

Gao, C., Liu, J., Kriebel, A.R., Preissl, S., Luo, C., Castanon, R., Sandoval, J., Rivkin, A., Nery, J.R., Behrens, M.M., *et al.* (2021). Iterative single-cell multi-omic integration using online learning. *Nat Biotechnol* 39, 1000-1007.

Gayoso, A., Lopez, R., Xing, G., Boyeau, P., Wu, K., Jayasuriya, M., Melhman, E., Langevin, M., Liu, Y., Samaran, J., *et al.* (2021a). scvi-tools: a library for deep probabilistic analysis of single-cell omics data. bioRxiv, 2021.2004.2028.441833.

Gayoso, A., Steier, Z., Lopez, R., Regier, J., Nazor, K.L., Streets, A., and Yosef, N. (2021b). Joint probabilistic modeling of single-cell multi-omic data with totalVI. *Nat Methods* 18, 272-282.

Hao, Y., Hao, S., Andersen-Nissen, E., Mauck, W.M., 3rd, Zheng, S., Butler, A., Lee, M.J., Wilk, A.J., Darby, C., Zager, M., *et al.* (2021). Integrated analysis of multimodal single-cell data. *Cell*.

Kingma, D.P., and Welling, M. (2013). Auto-Encoding Variational Bayes. arXiv:1312.6114.

Kotronen, A., Seppanen-Laakso, T., Westerbacka, J., Kiviluoto, T., Arola, J., Ruskeepaa, A.L., Oresic, M., and Yki-Jarvinen, H. (2009). Hepatic stearoyl-CoA desaturase (SCD)-1 activity and diacylglycerol but not ceramide concentrations are increased in the nonalcoholic human fatty liver. *Diabetes* 58, 203-208.

Lin, Y., Wu, T.Y., Wan, S., Yang, J.Y.H., Wong, W.H., and Wang, Y.X.R. (2022). scJoint integrates atlas-scale single-cell RNA-seq and ATAC-seq data with transfer learning. *Nat Biotechnol*.

Lopez, R., Regier, J., Cole, M.B., Jordan, M.I., and Yosef, N. (2018). Deep generative modeling for single-cell transcriptomics. *Nat Methods* 15, 1053-1058.

Lotfollahi, M., Naghipourfar, M., Luecken, M.D., Khajavi, M., Büttner, M., Avsec, Z., Misharin, A.V., and Theis, F.J. (2020). Query to reference single-cell integration with transfer learning. bioRxiv,

2020.2007.2016.205997.

Luecken, M.D., Buttner, M., Chaichoompu, K., Danese, A., Interlandi, M., Mueller, M.F., Strobl, D.C., Zappia, L., Dugas, M., Colome-Tatche, M., *et al.* (2022). Benchmarking atlas-level data integration in single-cell genomics. *Nat Methods* 19, 41-50.

Oetjen, K.A., Lindblad, K.E., Goswami, M., Gui, G., Dagur, P.K., Lai, C., Dillon, L.W., McCoy, J.P., and Hourigan, C.S. (2018). Human bone marrow assessment by single-cell RNA sequencing, mass cytometry, and flow cytometry. *JCI Insight* 3.

Schulte-Schrepping, J., Reusch, N., Paclik, D., Bassler, K., Schlickeiser, S., Zhang, B., Kramer, B., Krammer, T., Brumhard, S., Bonaguro, L., *et al.* (2020). Severe COVID-19 Is Marked by a Dysregulated Myeloid Cell Compartment. *Cell* 182, 1419-1440 e1423.

Veres, A., Faust, A.L., Bushnell, H.L., Engquist, E.N., Kenty, J.H., Harb, G., Poh, Y.C., Sintov, E., Gurtler, M., Pagliuca, F.W., *et al.* (2019). Charting cellular identity during human in vitro beta-cell differentiation. *Nature* 569, 368-373.

Wu, F.X., Wang, Q., Zhang, Z.M., Huang, S., Yuan, W.P., Liu, J.Y., Ban, K.C., and Zhao, Y.N. (2009). Identifying serological biomarkers of hepatocellular carcinoma using surface-enhanced laser desorption/ionization-time-of-flight mass spectroscopy. *Cancer Lett* 279, 163-170.

Xiong, L., Xu, K., Tian, K., Shao, Y., Tang, L., Gao, G., Zhang, M., Jiang, T., and Zhang, Q.C. (2019). SCALE method for single-cell ATAC-seq analysis via latent feature extraction. *Nat Commun* 10, 4576.

Zhang, J.Y., Wang, X.M., Xing, X., Xu, Z., Zhang, C., Song, J.W., Fan, X., Xia, P., Fu, J.L., Wang, S.Y., *et al.* (2020). Single-cell landscape of immunological responses in patients with COVID-19. *Nat Immunol* 21, 1107-1118.

Ziffra, R.S., Kim, C.N., Ross, J.M., Wilfert, A., Turner, T.N., Haeussler, M., Casella, A.M., Przytycki, P.F., Keough, K.C., Shin, D., *et al.* (2021). Single-cell epigenomics reveals mechanisms of human cortical development. *Nature* 598, 205-213.

REVIEWER COMMENTS

Reviewer #1 (Remarks to the Author):

The authors have addressed our concerns well. Overall, they provided substantial results to support their claims. We appreciate the clarification about how this method performs (without retraining) in contrast to continually updated online learning.

We have only a few small remaining concerns (which could be addressed without re-review):

1) While the authors now show results from scJoint and bindSC, they do not describe in the methods section how they used these tools, even though other comparison tool implementations are described.

2) Since this study involved different types of data analyses and benchmarking with many integration methods, it could be helpful if example code to reproduce major results in this manuscript were included in the github site.

Reviewer #2 (Remarks to the Author):

In their previous submission the authors had provided a method to integrate large scale datasets without requiring retraining

and large processing times and computational power. As a reviewer and based on my personal expertise, I had posed numerous questions regarding the biological significance and applicability in different contexts of this method. After revision, the authors have addressed all my concerns by either elaborating on the missing aspects or by carrying out additional analysis.

I am satisfied with their rebuttal and deem this study to be of interest for communities where such data integration is a major part of the experimental setup.

Reviewer #3 (Remarks to the Author):

I appreciate that the authors have aimed to address some of the data-specific issues that were brought up (such as the issue of inappropriate mixing in certain datasets) by tweaking parameters and visualizations; however, ultimately the key questions regarding the novelty of this method and whether it is actually superior to existing methods (including a very similar version of their own previously published one) has not been explicitly demonstrated. One of the main concerns I mentioned in my last review was regarding the novelty of this method compared to the author's own previous methods. I questioned whether the current updated model, in principle/in theory, can actually provide such drastic improvements compared to the previous model. They have verbally discussed some advantages of the new method, but in our discussion thus far, this represents a disagreement in opinion between myself and the authors, as both of us simple present verbal evidence. Thus, the onus is on the authors to do the benchmarking to prove how this updated model vastly improves performance. But they presented no such quantitative evidence, nor did they do any simulations or calculations to show that the new addition to the model actually improves the performance.

There have been several benchmarking papers published recently that do routine, systematic comparisons between single-cell integration methodologies, using now-standardized metrics to compare different methodologies (e.g. Luecken et al. 2022, <https://doi.org/10.1038/s41592-021-01336-8>). At the minimum the authors should follow this kind of rigorous benchmarking for their work, and do a quantitative comparison between not only other state-of-the-art methods, but also between this updated model and their own previously published model to show that indeed their updated model provides true measurable benefits/improvements that are not simply incremental.

Unfortunately, without such rigorous benchmarking, I cannot support this manuscript for publication in this form.

Reviewer #4 (Remarks to the Author):

Overall, the authors have worked diligently to address the majority of my concerns. The one aspect of my review that wasn't fully explored was the question of the impact of online integration (projection) in the context of the melanoma and PBMC datasets. The authors demonstrate that the method is robust to the order of projection of the melanoma datasets and show that melanoma specific cell types remained distinct after projection. I would be interested to examine to what degree information is lost in the projection process (are there melanoma specific signals that are being obscured) and to what degree is that quantifiable/reportable to the user when SCALEX is applied in this manner to study perturbed systems. In general though, this is a minor concern and is partially addressed earlier in the paper in the "over-correction scoring" sections. I feel comfortable recommending publication.

Reviewer #5 (Remarks to the Author):

The authors have well addressed the reviewer's concerns and it's suggested for publication.

Reviewer #6 (Remarks to the Author):

The authors performed comprehensive benchmark tests with existing methods, and provided substantial results to support their claims. They have addressed my previous concerns with new data analysis outcomes. Overall, I think this manuscript presents a new computational integration tool that can compete with other well-established methods and can attract a number of users.

The last concern would be code availability. It would be fair for the community to see data analysis code of other integration methods the authors used for all benchmark tests. In the method section, the authors didn't provide how they performed integration with scJoint and bindSC.

REVIEWER COMMENTS

Reviewer #1 (Remarks to the Author):

The authors have addressed our concerns well. Overall, they provided substantial results to support their claims. We appreciate the clarification about how this method performs (without retraining) in contrast to continually updated online learning.

We have only a few small remaining concerns (which could be addressed without re-review):

1) While the authors now show results from scJoint and bindSC, they do not describe in the methods section how they used these tools, even though other comparison tool implementations are described.

RESPONSE: We are sorry for not including the description of how we used scJoint and bindSC in the previous version of the manuscript. We have now added it in the methods section (see page 33 line 896~906), which reads as:

“scJoint: We used the scJoint Python package. We pre-processed the data into the standard input format for scJoint, and then modified the config.py file in the scJoint package and set the same training config parameters as used in the tutorial of “Analysis of PBMC data from 10x Genomics using scJoint” (<https://github.com/sydneybio/scJoint/blob/main/tutorial/Analysis%20of%2010xGenomics%20data%20using%20scJoint.ipynb>).

bindSC (v1.0.0): We used the bindSC R package. Following the tutorial, we first performed dimension reductions for gene expression, for the gene activity scores, and for the chromatin accessibility profiles, using the dimReduce function with K=30. Subsequently, we ran the BiCCA function with lambda=0.5, alpha=0.5, and K=20. All other parameters were default.”

2) Since this study involved different types of data analyses and benchmarking with many integration methods, it could be helpful if example code to reproduce major results in this manuscript were included in the github site.

RESPONSE: We have now uploaded our analysis code to GitHub: <https://github.com/jsxlei/SCALEX>.

We would like to take this opportunity to express our gratitude to the review for the excellent guidance about how to improve our study and manuscript.

Reviewer #2 (Remarks to the Author):

In their previous submission the authors had provided a method to integrate large scale datasets without requiring retraining and large processing times and computational power. As a reviewer and based on my personal expertise, I had posed numerous questions regarding the biological significance and applicability in different contexts of this method. After revision, the authors have addressed all my concerns by either elaborating on the missing aspects or by carrying out additional analysis.

I am satisfied with their rebuttal and deem this study to be of interest for communities where such data integration is a major part of the experimental setup.

We really appreciate the review for the excellent guidance about how to improve our study regarding the biological significance and applicability. Many thanks.

Reviewer #3 (Remarks to the Author):

I appreciate that the authors have aimed to address some of the data-specific issues that were brought up (such as the issue of inappropriate mixing in certain datasets) by tweaking parameters and visualizations; however, ultimately the key questions regarding the novelty of this method and whether it is actually superior to existing methods (including a very similar version of their own previously published one) has not been explicitly demonstrated. One of the main concerns I mentioned in my last review was regarding the novelty of this method compared to the author's own previous methods. I questioned whether the current updated model, in principle/in theory, can actually provide such drastic improvements compared to the previous model. They have verbally discussed some advantages of the new method, but in our discussion thus far, this represents a disagreement in opinion between myself and the authors, as both of us simple present verbal evidence. Thus, the onus is on the authors to do the benchmarking to prove how this updated model vastly improves performance. But they presented no such quantitative evidence, nor did they do any simulations or calculations to show that the new addition to the model actually improves the performance.

There have been several benchmarking papers published recently that do routine, systematic comparisons between single-cell integration methodologies, using now-standardized metrics to compare different methodologies (e.g. Luecken et al.

2022, <https://doi.org/10.1038/s41592-021-01336-8>). At the minimum the authors should follow this kind of rigorous benchmarking for their work, and do a quantitative comparison between not only other state-of-the-art methods, but also between this updated model and their own previously published model to show that indeed their updated model provides true measurable benefits/improvements that are not simply incremental.

Unfortunately, without such rigorous benchmarking, I cannot support this manuscript for publication in this form.

RESPONSE: We appreciate the reviewer for the guidance again, and we have now tested the integration performance of SCALE (Xiong et al., 2019) on the benchmark datasets. Very briefly, based on these “quantitative evidence” from the benchmarking experiments, we do see that the new SCALEX model vastly outperforms SCALE for single-cell data integration (**Revision Figures 1 and 2, Revision Table 1**).

Revision Figure 1. Integration performance of SCALE on the benchmark datasets. UMAP embeddings of the benchmark datasets after integration by SCALE. Cells are colored by batch or cell-type.

Revision Figure 2. Systematic comparisons of integration performance by the now-standardized metrics. Dotplot shows the scores and rankings of different methods on different metrics across benchmark datasets. The scores for all 12 examined metrics (circles) were calculated using the Python package *scIB* (Luecken et al., 2022) with default parameters. Note that due to the benchmark datasets aren't developmental datasets and some methods don't return a corrected data matrix (cell x gene) after integration, we didn't include the trajectory conservation score or the HVG conservation score in the comparison. The `batch_correction_mean`, `bio_conservation_mean`, and `overall` scores (rectangles) were calculated as described in (Luecken et al., 2022) to assess the performances of different methods in terms of the batch removal, the conservation of biological variance, and overall accuracy scores, respectively.

Revision Table 1. Comparison of SCALEX and SCALE using the metrics from Luecken et al. 2022 on benchmark datasets.

		PCR_b atch	ASW_I abel/b atch	iLISI	graph_ conn	kBET	NMI_cl uster/l abel	ARI_cl uster/l abel	ASW_I abel	isolate d_labe l_F1	isolate d_labe l_silho uette	cLISI	cell_cy cle_co nserva tion	batch_ correct ion_m ean	bio_co nserva tion_m ean	overall score
pancreas	SCALE	0.000	0.716	1.402	0.750	0.000	0.534	0.226	0.479	0.201	0.749	0.766	0.454	0.062	0.196	0.143
	SCALEX	0.998	0.879	3.625	0.992	0.270	0.926	0.908	0.615	0.407	0.584	0.916	0.500	0.850	0.733	0.780
	Δ	0.998	0.163	2.223	0.242	0.270	0.392	0.682	0.136	0.206	-0.165	0.150	0.046	0.788	0.537	0.637
liver	SCALE	0.000	0.569	1.001	0.841	0.000	0.713	0.563	0.564	0.611	0.531	0.773	0.624	0.093	0.160	0.133
	SCALEX	0.985	0.856	1.343	1.000	0.273	0.929	0.914	0.604	0.927	0.538	0.984	0.727	0.810	0.757	0.778
	Δ	0.985	0.287	0.342	0.159	0.273	0.216	0.351	0.040	0.316	0.007	0.211	0.103	0.717	0.597	0.645
heart	SCALE	0.000	0.702	1.050	0.981	0.017	0.776	0.612	0.604	0.403	0.537	0.942	0.297	0.247	0.309	0.284
	SCALEX	0.984	0.836	1.527	0.988	0.397	0.925	0.960	0.665	0.894	0.569	0.962	0.298	0.808	0.713	0.751
	Δ	0.984	0.134	0.477	0.007	0.380	0.149	0.348	0.061	0.491	0.032	0.020	0.001	0.561	0.404	0.467
NSCLC	SCALE	0.000	0.769	1.298	0.962	0.299	0.633	0.362	0.494	0.680	0.448	0.645	0.242	0.287	0.073	0.158
	SCALEX	0.997	0.885	1.582	0.997	0.622	0.880	0.744	0.617	0.810	0.530	0.952	0.478	0.876	0.763	0.808
	Δ	0.997	0.116	0.284	0.035	0.323	0.247	0.382	0.123	0.130	0.082	0.307	0.236	0.589	0.690	0.650
PBMC	SCALE	0.000	0.600	1.000	0.706	0.000	0.616	0.331	0.505	0.524	0.517	0.810	0.616	0.00	0.113	0.068
	SCALEX	1.000	0.920	1.734	0.989	0.620	0.923	0.936	0.632	0.895	0.583	0.951	0.253	0.886	0.715	0.783
	Δ	1.000	0.320	0.734	0.283	0.620	0.307	0.605	0.127	0.371	0.066	0.141	-0.363	0.886	0.602	0.715

The fact that SCALEX consistently, dramatically outperformed SCALE is not surprising when considering the basic designs of these tools. SCALE was designed to be a dimension reduction and data imputation tool for individual datasets; that is, it was not designed for the purpose of integrating different datasets. We want to emphasize that the now clearly demonstrated superior data integration performance of SCALEX (**Revision Table 1**) comes from its design of an asymmetric autoencoder that inputs batch information only to the decoder, a DSBN layer in the decoder to release the encoder from the burden of capturing the batch-specific variations, and a mini-batching strategy that samples data from all batches instead from a single batch and thus more tightly follows the same overall distribution of the input data; this strategy includes a Batch Normalization layer in the encoder that adjusts the deviation of each mini-batch and align them to the overall input distribution. Our ablation study has highlighted the contributions of three design elements of SCALEX on removing batch effects. None of these designs are in the original SCALE model.

We have also followed the reviewer's guidance and benchmarked SCALEX against other state-of-the-art methods (as well as SCALE, described above) using the now-standardized metrics (Luecken et al., 2022) (**Revision Figure 2**). Indeed, SCALEX outperformed all other state-of-the-art single-cell data integration tools on *pancreas*, *liver*, and *NSCLC* datasets in terms of the overall score, and ranked the third on *PBMC* and the fourth on *heart*. Note that SCALE was the bottom-ranking for all five datasets in terms of the overall score.

Reviewer #4 (Remarks to the Author):

Overall, the authors have worked diligently to address the majority of my concerns. The one aspect of my review that wasn't fully explored was the question of the impact of online integration (projection) in the context of the melanoma and PBMC datasets. The authors demonstrate that the method is robust to the order of projection of the melanoma datasets and show that melanoma specific cell types remained distinct after projection. I would be interested to examine to what degree information is lost in the projection process (are there melanoma specific signals that are being obscured) and to what degree is that quantifiable/reportable to the user when SCALEX is applied in this manner to study perturbed systems. In general though, this is a minor concern and is partially addressed earlier in the paper in the "over-correction scoring" sections. I feel comfortable recommending publication.

RESPONSE: We appreciate the reviewer's ongoing support of our efforts in this study. We are also deeply interested in the performance of SCALEX as applied to perturbed systems, and anticipate that the fundamental utility of SCALEX for users will indeed be based on its capacity to retain informative signals for specific cell types. It will be exciting to see how SCALEX performs in our ongoing collaborative studies and ideally upon its application by the wider research community.

Reviewer #5 (Remarks to the Author):

The authors have well addressed the reviewer's concerns and it's suggested for publication.

We would like to take this opportunity to express our gratitude to the review for the excellent guidance about how to improve our study.

Reviewer #6 (Remarks to the Author):

The authors performed comprehensive benchmark tests with existing methods, and provided substantial results to support their claims. They have addressed my previous concerns with new data analysis outcomes. Overall, I think this manuscript presents a new computational integration tool that can compete with other well-established methods and can attract a number of users.

The last concern would be code availability. It would be fair for the community to see data analysis code of other integration methods the authors used for all benchmark tests. In the method section, the authors didn't provide how they performed integration with scJoint and bindSC.

RESPONSE: We thank the reviewer for these suggestions, and we have uploaded our analysis code to GitHub: <https://github.com/jsxlei/SCALEX>.

Additionally, we have now added the description of how we used scJoint and bindSC in the methods section (see page 33 line 896~906), which reads as:

"scJoint: We used the scJoint Python package. We pre-processed the data into the standard input format for scJoint, and then modified the config.py file in the scJoint package and set the same training config parameters as used in the tutorial of "Analysis of PBMC data from 10x Genomics using scJoint" (<https://github.com/sydneybiox/scJoint/blob/main/tutorial/Analysis%20of%201>

0xGenomics%20data%20using%20scJoint.ipynb).

bindSC (v1.0.0): We used the bindSC R package. Following the tutorial, we first performed dimension reductions for gene expression, for the gene activity scores, and for the chromatin accessibility profiles, using the dimReduce function with K=30. Subsequently, we ran the BiCCA function with lambda=0.5, alpha=0.5, and K=20. All other parameters were default.”

References

- Luecken, M.D., Buttner, M., Chaichoompu, K., Danese, A., Interlandi, M., Mueller, M.F., Strobl, D.C., Zappia, L., Dugas, M., Colome-Tatche, M., *et al.* (2022). Benchmarking atlas-level data integration in single-cell genomics. *Nat Methods* 19, 41-50.
- Xiong, L., Xu, K., Tian, K., Shao, Y., Tang, L., Gao, G., Zhang, M., Jiang, T., and Zhang, Q.C. (2019). SCALE method for single-cell ATAC-seq analysis via latent feature extraction. *Nat Commun* 10, 4576.

REVIEWERS' COMMENTS

Reviewer #3 (Remarks to the Author):

Thanks for taking my review into consideration and generating the necessary benchmarking data. I can recommend publication if the authors include the new figures of the benchmarking results in the supplemental figures.

REVIEWER COMMENTS

Reviewer #3 (Remarks to the Author):

Thanks for taking my review into consideration and generating the necessary benchmarking data. I can recommend publication if the authors include the new figures of the benchmarking results in the supplemental figures.

RESPONSE: We have now added the new figures of the benchmarking results in Supplementary Fig. 4b. We thank the reviewer for continuous help and support.